# The ORC ubiquitin ligase OBI1 promotes DNA replication origin firing

Philippe Coulombe[1,4], Joelle Nassar [1,4], Isabelle Peiffer[1], Slavica Stanojcic[2], Yvon Sterkers[2,3], Axel Delamarre[1], Stéphane Bocquet[1] & Marcel Méchali[1]

DNA replication initiation is a two-step process. During the G1-phase of the cell cycle, the ORC complex, CDC6, CDT1, and MCM2–7 assemble at replication origins, forming pre-replicative complexes (pre-RCs). In S-phase, kinase activities allow fork establishment through (CDC45/MCM2–7/GINS) CMG-complex formation. However, only a subset of all potential origins becomes activated, through a poorly understood selection mechanism. Here we analyse the pre-RC proteomic interactome in human cells and find C13ORF7/RNF219 (hereafter called OBI1, for ORC-ubiquitin-ligase-1) associated with the ORC complex. OBI1 silencing result in defective origin firing, as shown by reduced CMG formation, without affecting pre-RC establishment. OBI1 catalyses the multi-mono-ubiquitylation of a subset of chromatin-bound ORC3 and ORC5 during S-phase. Importantly, expression of non-ubiquitylable ORC3/5 mutants impairs origin firing, demonstrating their relevance as OBI1 substrates for origin firing. Our results identify a ubiquitin signalling pathway involved in origin activation and provide a candidate protein for selecting the origins to be fired.

[1] Institute of Human Genetics, UMR 9002, CNRS-Université de Montpellier, 141 rue de la Cardonille, 34396 Montpellier, France. [2] CNRS 5290 - IRD 224 - University of Montpellier (UMR "MiVEGEC"), 34090 Montpellier, France. [3] University Hospital Centre (CHU), Department of Parasitology-Mycology, 34090 Montpellier, France. [4] These authors contributed equally: Philippe Coulombe, Joelle Nassar. Correspondence and requests for materials should be addressed to P.C. (email: philippe.coulombe@igh.cnrs.fr) or to M.M. (email: marcel.mechali@igh.cnrs.fr)

DNA replication is initiated at defined genomic sites called origins of replication (see refs. [1,2] for review). Potential DNA replication origin sites (hereafter called origins) are established during the G1 phase of the cell cycle through a highly regulated process known as replication licensing[3,4] (for review). In eukaryotes, origins are recognised by the ORC complex, a multi-subunit (ORC1–6 and LRWD1/ORCA in vertebrates) AAA + ATPase[5]. After ORC binding, CDC6 and CDT1 assemble at origins and permit the licensing reaction that culminates with the loading onto chromatin of the replicative helicase MCM2/7 in its inactive state, defining the pre-replicative complexes (pre-RCs). Although ORC binding is sequence-specific in *S. cerevisiae*, it is unclear how ORC recognises origins in metazoans.

As cells enter S phase, the MCM2/7 helicase is activated at pre-RCs through CDK (Cyclin-dependent kinase) and DDK (DBF4-dependent kinase) activities[6]. This leads to the association of CDC45 and the GINS complex to chromatin-bound MCM2/7 and, consequently, to the formation of the active CMG complex[7]. CMG catalyses the unwinding of the DNA, allowing origin firing and the establishment of active replication forks. In mammals, origins are relatively inefficient and only a fraction of pre-RCs (5–40% of all potential origins) are activated, in a flexible manner, during each cell cycle[8–10]. Moreover, the repertoire of origin usage can differ from cell to cell[11]. The mechanisms involved in selective activation of origins are poorly understood.

To gain further insights into origin establishment and activation, we characterised the human pre-RC interactome using a proteomic approach. In addition to already known interacting partners, we identified previously undescribed pre-RC associated proteins. Among these, we focused on C13ORF7/RNF219, an uncharacterised potential E3 ubiquitin ligase that we called OBI1 (ORC ubiquitin-ligase-1). The properties of OBI1 suggest that it could be a replication origin selector essential for DNA replication origin activation during S phase.

## Results

**The human pre-RC interactome.** To gain further insights into replication origin biology, we characterised the human pre-RC proteomic interactome. In HeLa S3 cell lines that stably express several pre-RC components (ORC1, ORC2, ORCA/LRWD1, CDC6 and CDT1) C-terminally tagged with Flag/HA. The expression levels of these baits and of the endogenous proteins were comparable (Supplementary Fig. 1a). After purification of baits and the associated proteins from Dignam nuclear extracts using a sequential immunoprecipitation/peptide elution approach (TAP-TAG), see Supplementary Fig. 1b, c and Methods), we analysed a fraction of the final eluates by gel electrophoresis/silver staining (Fig. 1a), and by mass spectrometry for protein identification (Fig. 1a, b and Supplementary Tables 1 and 2). Mass spectrometry identified previously known pre-RC interactors. Players and regulators of origin licensing and firing (ORC/ORCA complexes, CDKs, geminin, SCF^SKP2 and ubiquitin) as well as factors involved in chromatin transactions (G9a/b, shelterin complex, Ku proteins and histones) were amongst these known interactors (more details in Supplementary Discussion). Western blot analysis confirmed the presence of selected interactors in the purified complexes (Fig. 1c). Among the identified pre-RC interactors that were previously undescribed, we found factors associated with the nuclear matrix (RIF1, NUMA and AKAP8), RNA metabolism (DDX5, DDX39B, FXR1, HNRPM, RBM10 and PRMT5), transcription (NFIL3 and TRIM29), and the ubiquitin pathway (C13ORF7/RNF219 and PSME3).

**OBI1, an ORC complex binding protein.** This study focuses on C13ORF7/RNF219, a putative E3 ubiquitin ligase with unknown function(s). Based on the biochemical function of C13ORF7/RNF219 reported herein, we propose to name this enzyme OBI1, for ORC-ubiquitin-ligase-1. OBI1 has an N-terminal RING domain followed by a coiled-coil domain and a large C-terminal extension without any recognisable domain (Fig. 1d). We found homologues of this putative ubiquitin ligase in all metazoans, with the same overall architecture (Supplementary Fig. 2a). Phylogenetic analysis suggested that ecdysozoans, which include arthropods and nematodes, lost this enzyme during evolution. OBI1's RING domain showed the strongest conservation (Supplementary Fig. 2a, b).

Western blot analysis confirmed the presence of endogenous OBI1 in purified ORC complexes (ORC1, ORC2 and ORCA baits, Fig. 1c), but not with the CDC6 or CDT1 baits. Both endogenous and ectopic OBI1 were associated with the ORC complex in proliferating cells (Fig. 1e, Supplementary Figs. 3a–c and 10d). OBI1 association with various ORC complex subunits suggested that it interacts with the whole ORC complex. To characterise OBI1 domain(s) required for ORC binding, we generated OBI1 deletion mutants and assessed their association with ORC1. OBI1 C-terminal region and coiled-coil domain were both required for ORC interaction (Supplementary Fig. 3a), but not the RING domain. Moreover, a catalytically inactive OBI1 mutant (Cys38Ser, see Supplementary Fig. 2b, further details later) interacted with the ORC complex (Supplementary Fig. 3d), showing that OBI1 ubiquitin ligase activity is not required for ORC association.

We then monitored the expression and chromatin association of endogenous OBI1 in U2OS cells synchronised in mitosis and then released in the cell cycle. Western blot analysis revealed that OBI1 expression was fairly constant throughout the cell cycle, while its chromatin association was cell cycle-regulated (Fig. 1f). OBI1 was absent from mitotic chromosomes, but was associated with chromatin from G1, and partially released from chromatin from mid S-phase. Moreover, in mitotic cell extracts, OBI1 band showed a molecular weight shift (Fig. 1f, left panel), suggesting a possible cell cycle regulation through phosphorylation, as documented for several replication origin factors[4].

**OBI1 is involved in cell growth and transformation.** Proper DNA replication is needed for cell growth. Indeed, siRNA-mediated silencing of ORC1 and CDC7, which are essential factors for origin licensing and firing respectively, impaired cell proliferation (Fig. 2a, siORC1- and siCDC7-transfected cells), compared with cells transfected with a non-targeting siRNA (siMock, see Supplementary Table 3 for sequences). Similarly, *OBI1* knockdown using siRNA pools targeting the coding sequence (siOBI1) or the 3′UTR (siUTR) significantly reduced cell proliferation (Fig. 2a). As OBI1 is a positive cell growth regulator, we evaluated its expression in human cancer samples using the ONCOMINE server[12]. OBI1 was overexpressed in different tumours, particularly colorectal adenocarcinoma (Supplementary Fig. 4a). We then investigated OBI1 potential oncogenic properties using classical transformation assays in non-transformed mouse NIH 3T3 cells[13]. OBI1 overexpression abrogated contact inhibition and allowed anchorage-independent cell growth (Supplementary Fig. 4b–d), two hallmarks of cell transformation. In these conditions, control NIH 3T3 cells did not form foci at confluence and colonies in soft-agar.

**OBI1 is essential for origin firing but not for origin licensing.** We then assessed whether OBI1 was involved in DNA replication. Cell cycle profile analysis by flow cytometry of siRNA-transfected U2OS cells pulsed with the thymidine analogue BrdU showed that *OBI1* knockdown (siOBI1 and siUTR) resulted in a

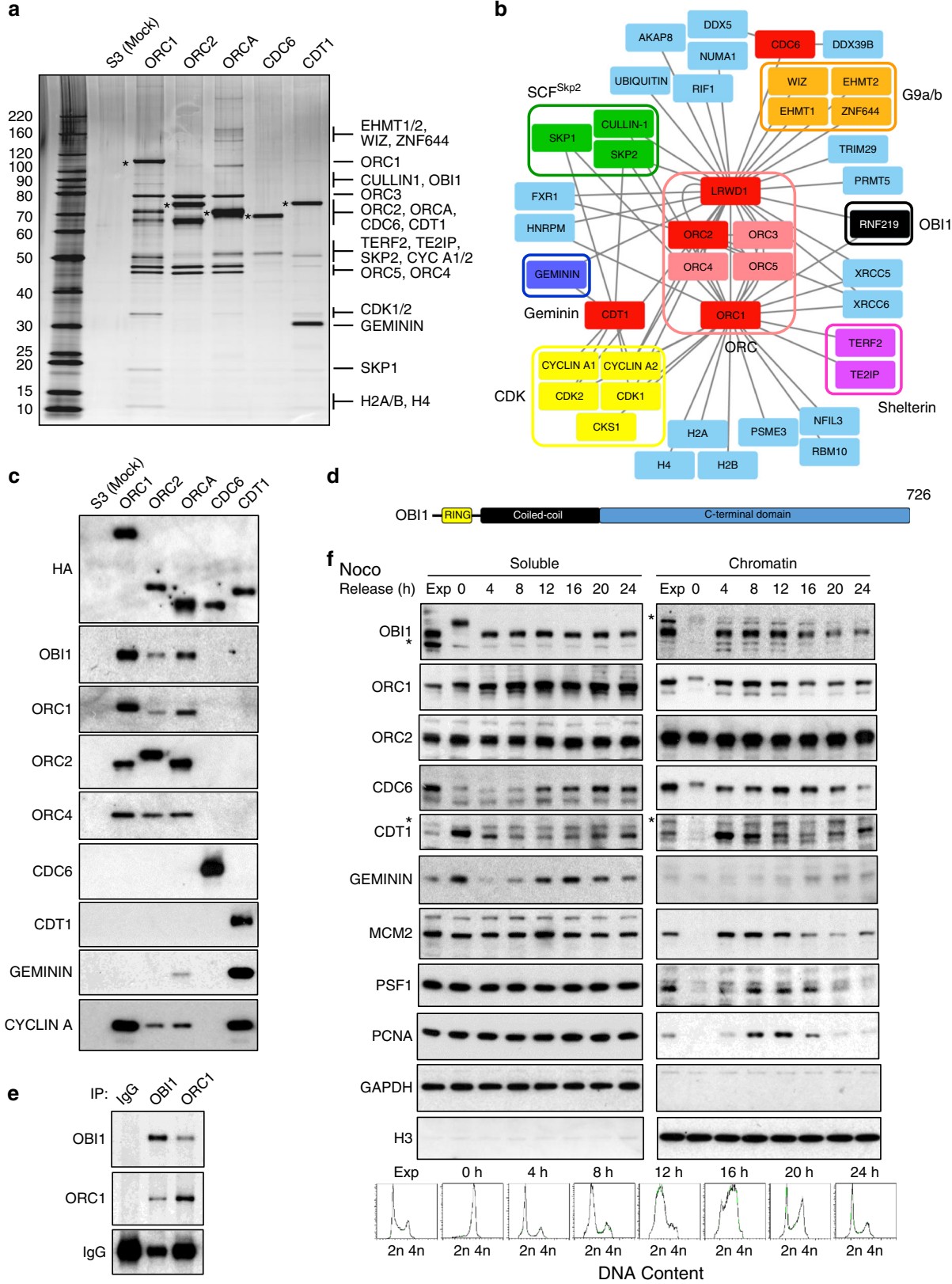

noticeable accumulation of cells in the S- and G2/M-phases, compared with control (siMock, Fig. 2b). BrdU incorporation per cell, reflecting the overall DNA synthesis, was reduced by ~50% in *OBI1* knockdown cells (BrdU fluorescence intensity quantified by flow cytometry) (Fig. 2b, right panel). *OBI1* knockdown led to a similar DNA synthesis defect also in HCT 116 and T98G cells

(Supplementary Fig. 5). ORC1 and CDC7 depletion, which decreases the number of licensed and fired origins, respectively, led to a similar reduction of BrdU fluorescence intensity level (Fig. 2b). Silencing of treslin or its associated protein MTBP, which are essential components of origin firing, also caused a similar DNA synthesis defect[14,15].

**Fig. 1** The pre-RC interactome analysis identifies OBI1 (C13ORF7/RNF219) as an ORC-associated protein. **a** The indicated baits were TAP-TAG purified from Dignam nuclear extracts of stably expressing HeLa S3 cell lines. Then, 20% of the final eluates were resolved on gradient acrylamide gels and visualised by silver staining. Asterisks mark the Flag-HA-tagged baits. The molecular weight markers are indicated in kDa. **b** Schematic representation of the human pre-RC interactome. Baits used in this study are marked in red. Known pre-RC interactors are highlighted in different colours, while novel binders are in light blue. OBI1 (C13ORF7/RNF219) is highlighted in black. Interactors are grouped based on their known interactions. **c** The TAP-TAG final eluates were analysed by western blotting using antibodies against the indicated proteins. **d** Schematic representation of human OBI1 with the RING, coiled-coil and C-terminal domains highlighted. **e** Immunoprecipitation/western blotting analysis confirmed the association of endogenous OBI1 and ORC1 in HeLa S3 cell Dignam extracts. **f** Cell cycle regulation of OBI1 expression and chromatin association. Exponentially growing U2OS cells (Exp) were synchronised in mitosis by incubation with nocodazole and then released in normal growth medium for the indicated times. Soluble and chromatin fractions were analysed by western blotting with antibodies against the indicated proteins. Synchronisation was evaluated by flow cytometry (lower panels). Asterisks mark non-specific bands

To further characterise the DNA synthesis defects induced by *OBI1* silencing, we studied DNA replication dynamics using DNA combing and DNA stretching assays (Fig. 2c–f and Supplementary Fig. 6a, b, respectively; see Methods). It is well established that reducing DNA replication initiation events results in higher replication fork speed, as a compensatory mechanism[15–19]. In agreement, fork speed was increased upon ORC1 or CDC7 depletion (Fig. 2c, d), as previously observed[17], and also upon *OBI1* silencing (Fig. 2c, d). Quantification of the origin firing rate by measuring the Inter-Origin Distances (IOD) and Global Fork Densities (GFD) in silenced cells showed that the reduction in DNA synthesis initiation events, as expected in *ORC1* and *CDC7* knockdown cells, led to larger IOD and lower GFD values (Fig. 2e, f), as previously reported[16,17]. *OBI1* silencing also resulted in higher IOD and lower GFD values compared with control cells (Fig. 2e, f). Similarly, DNA fibre stretching analysis showed replication fork acceleration, consistent with the activation defect associated with *OBI1* silencing (Supplementary Fig. 6a, b).

Then, we analysed the expression and chromatin association of key replication factors to determine which step of replication origin activity (origin licensing or firing) is regulated by OBI1. Upon ORC1 depletion, recruitment of the MCM2/7 complex to chromatin was impaired (Fig. 2g, Supplementary Figs. 6c and 11), confirming defective origin licensing. Consequently, chromatin binding of factors involved in the subsequent firing step, such as DNA polymerase-α, RPA32, PSF1/GINS1, the processivity factor PCNA and its loader RFC2, also was reduced (Fig. 2g, Supplementary Figs. 6c and 11). On the other hand, *OBI1* knockdown did not affect ORC1, ORC2, ORC4, CDC6 or MCM2/7 recruitment to chromatin (Fig. 2g, Supplementary Figs. 6c, d and 11), demonstrating that OBI1 is not required for licensing. However, chromatin recruitment of replisome components, such as DNA polymerase α, RPA32, PSF1/GINS1, PCNA and RFC2, was impaired by ~50% (Fig. 2g, Supplementary Figs. 6d and 11). We observed a similar phenotype upon silencing of the firing factor CDC7 (Fig. 2g and Supplementary Fig. 11), consistent with impaired CMG complex formation. The observed phenotype was not a consequence of inappropriate activation of the DNA damage checkpoint, as indicated by the absence of CHK1 phosphorylation or CHK1 induction upon OBI1 knockdown (Supplementary Fig. 6c). Altogether, these findings indicate that OBI1 is critical for origin firing, but not for origin licensing.

**Chromatin-bound ORC3 and ORC5 are multi-mono-ubiquitylated in a cell cycle regulated manner.** As OBI1 is an ORC interactor and a predicted ubiquitin ligase, it could positively regulate origin firing by direct ubiquitylation of the ORC complex. To test this hypothesis, we first assessed the ubiquitylation status of the ORC complex in vivo (see Methods). In these experiments, we co-expressed each tagged ORC subunit with HA-ubiquitin, and analysed their ubiquitylation status by western blotting. We detected ORC1 ubiquitylation, as previously

described[20,21], while ORC2, ORC4 and ORC6 ubiquitylation was found to be very low (Fig. 3a and Supplementary Fig. 7 for controls). Strikingly, we found that ORC3 and ORC5 subunits (ORC3/5) were strongly ubiquitylated in vivo (Fig. 3a). In agreement, published large-scale proteomic studies to identify ubiquitylated proteins found many ubiquitylation sites on endogenous ORC3/5 in human cells[22,23] (see Supplementary Fig. 10a).

We further studied this previously uncharacterised post-translational modification of the ORC complex. We found that ORC3/5 purified from cellular fractions under denaturing conditions were ubiquitylated mainly on chromatin (Fig. 3b). Then, we studied the linkage type of ubiquitin assembled on ORC subunits using different approaches. Indeed, besides being covalently attached to substrates (known as mono-ubiquitylation), ubiquitin can be linked to any of its seven internal lysine residues (poly-ubiquitylation), yielding different outcomes depending on the nature of the ubiquitin linkage type[24,25]. For example, lysine 48-linked poly-ubiquitin chains target for proteasomal degradation. First, we co-expressed ORC3/5 with single-lysine ubiquitin mutants and analysed their ubiquitylation status. These ubiquitin mutants are proficient for a single linkage type poly-ubiquitylation, and can also be attached to substrates, yielding mono-ubiquitylation. All seven single-lysine ubiquitin mutants ubiquitylated ORC3/5, like wild type (WT) ubiquitin (Fig. 3c and Supplementary Fig. 8a). Moreover, lysine-less ubiquitin (0 K), which cannot sustain poly-ubiquitylation, but is still proficient for mono-ubiquitylation, was efficiently linked to ORC3/5 (Fig. 3c and Supplementary Fig. 8a). These results suggested that ORC3/5 are modified by mono-ubiquitylation. This conclusion was also supported by the results obtained with the UbiCRest assay, based on treatment of ubiquitylated substrates with linkage-specific deubiquitinating enzymes (DUBs)[26]. The universal DUB USP2$_{CD}$, which can cleave any kind of ubiquitin attachment, efficiently eliminated high molecular weight species of immunoprecipitated ORC3/5, releasing free ubiquitin in the supernatant (Fig. 3d and Supplementary Fig. 8b), illustrating again that these proteins are ubiquitylated in vivo. Conversely, ubiquitin linkage-specific DUBs did not affect ORC3/5 ubiquitylation (Fig. 3d and Supplementary Fig. 8b). Nevertheless, linkage-specific DUBs displayed activity in extracts supplemented with HA-ubiquitin that showed various types of ubiquitylation (Supplementary Fig. 8c), indicating that linkage-specific DUBs were active. The apparent molecular weight of ubiquitylated ORC3/5 species suggested that many ubiquitin molecules (2 to ~20) are conjugated to these substrates. This modification is referred to as multi-mono-ubiquitylation (or multi-ubiquitylation)[24], and is believed to have signalling roles, for instance in regulating protein/protein interactions[25].

Finally, to study the regulation of ORC3/5 ubiquitylation during the cell cycle, we generated stable U2OS cell lines that express tagged ORC3 or ORC5, and assessed ORC3/5 ubiquitylation and cell cycle profile at different times points after their

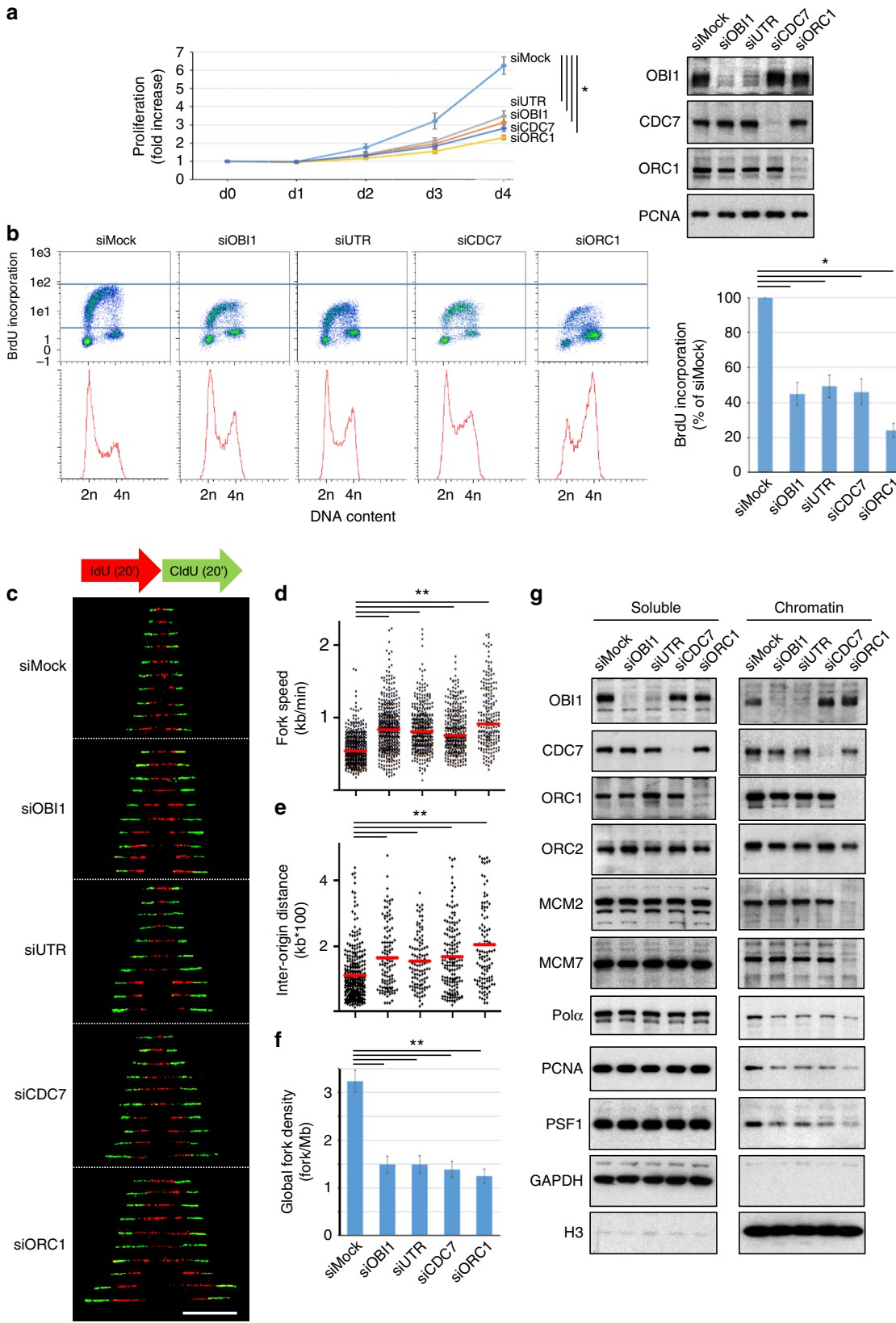

synchronisation in mitosis with nocodazole and release in fresh medium. ORC3/5 ubiquitylation, as observed by the appearance of high molecular weight species, was low in mitotic and early G1-phase cells, whereas it was induced in late G1-/early S-phase, peaked in S-phase, and then decreased toward the end of the cell cycle (Fig. 3e). Also, these experiments showed that only a subset of ORC3/5 became ubiquitylated (estimated to be 5–10%) during S-phase. Altogether, these results showed that chromatin-bound ORC3/5 become multi-mono-ubiquitylated concomitantly with origin firing.

**Fig. 2** OBI1 is required for replication origin firing. **a** Involvement of OBI1 in cell proliferation. U2OS cells were transfected with siRNA pools targeting OBI1 3′ UTR (siUTR) or coding sequence (siOBI1), ORC1 (siORC1), CDC7 (siCDC7) or a non-targeting siRNA (siMock) (sequences in Supplementary Table 3). Cell proliferation (fold-increase relative to day 0) was evaluated by counting cells every day after transfection. The mean results of three independent experiments are shown. Expression of endogenous OBI1, ORC1, CDC7 and PCNA was monitored by western blotting at day 3 (right). **b** U2OS cells were transfected with siRNAs as in **a**. Three days post-transfection, cells were incubated with BrdU for 15 min. BrdU incorporation and DNA content were analysed by flow cytometry (left panels). Lines delimiting BrdU-positive siMock-treated cells are shown. BrdU incorporation fluorescence signal was quantified from three independent experiments (right panel). **c** U2OS cells were transfected with siRNAs as in **a**. Three days post-transfection, cells were incubated with IdU (20 min) followed by CldU (20 min) and processed for DNA combing analysis (see Methods). Representative images of bidirectional forks labelling are shown. **d** Analysis of replication fork speed (in kb/min) in the cells described in **c**, based on the measurement of CldU tracks preceded by the IdU signal (two independent experiments). Red bars indicate median values. **e** Inter-origin distances (in kb) in the cells described in **c** were quantified from two independent experiments. Red bars indicate median values. **f** The mean global fork density (in fork/Mb) in the cells described in **c** was quantified by measuring the number of labelled forks per megabase of combed DNA, normalised to the percentage of S-phase cells (two independent experiments). **g** U2OS cells were transfected with siRNAs as in **a**. Three days later, chromatin and soluble fractions were isolated and analysed by western blotting with antibodies against the indicated proteins. *$p < 0.01$; **$p < 0.001$

**OBI1 catalyses ORC3 and ORC5 multi-mono-ubiquitylation.** We next assessed the possible involvement of OBI1 in ORC3/5 ubiquitylation using gain and loss of function experiments and in vitro assays. First, OBI1 overexpression stimulated mainly ORC3/5 ubiquitylation, leaving the other tested subunits largely unaffected (Fig. 4a). Although we cannot exclude that OBI1 might modify the other ORC subunits, we decided to focus on OBI1-induced ORC3/5 ubiquitylation because it was very robust. We observed this enhanced ubiquitylation when ORC3/5 were purified in native or in denaturing conditions (Fig. 4b–c and Supplementary Fig. 9a, b), showing that the modification is specific to these ORC subunits. OBI1-induced ORC3/5 ubiquitylation required its intrinsic ubiquitin ligase activity, because the OBI1 Cys38Ser (CS) and RING-deleted (ΔRING) mutants were inactive (Fig. 4b–c and Supplementary Figs. 2b and 9a–c). OBI1 mutants lacking the coiled-coil or the C-terminal extension (unable to bind to the ORC complex, see Supplementary Fig. 3a) also were inactive (Supplementary Fig. 9c). OBI1 ubiquitin ligase activity toward the ORC complex is a conserved feature because the *Xenopus laevis* OBI1 homologue also could catalyse human ORC5 ubiquitylation (Supplementary Fig. 9d). Importantly, *OBI1* siRNA-mediated knockdown in human U2OS cells drastically reduced ORC3/5 ubiquitylation (Fig. 4d–e). These findings identified OBI1 as a major ORC3/5 ubiquitin ligase in vivo. Nevertheless, we do not formally exclude the possibility that other E3 ligase(s) could be involved in ORC3/5 ubiquitylation, depending on the physiological and cellular context. Finally, we performed in vitro ubiquitylation assays to test whether OBI1 could directly modify ORC3/5. WT OBI1 but not the CS mutant purified from cell extracts ubiquitylated in vitro translated ORC3/5 (Fig. 4f–g), yielding a signal very similar to the one observed in cells. In these in vitro conditions, we found that OBI1 auto-ubiquitylates, as previously described for other E3 ligases[27,28]. In addition, in vitro assays using WT or 0 K ubiquitin gave very similar ORC3/5 ubiquitylation patterns (Fig. 4h), showing that OBI1 directly catalysed ORC3/5 multi-mono-ubiquitylation. Altogether, these results indicate that OBI1 is a major cellular ORC3/5 ubiquitin ligase capable of catalysing their multi-mono-ubiquitylation.

**ORC3 and ORC5 ubiquitylation is important for origin firing.** If ORC3/5 were functionally relevant OBI1 substrates for origin activation, then expression of non-ubiquitylable ORC mutants should mimic the effect of OBI1 depletion. Mutation of the lysine residues identified in large-scale ubiquitylome studies[22,23] (nine and seven lysine residues in ORC3 and ORC5, respectively, Supplementary Fig. 10a) into arginine to hinder ubiquitin attachment did not affect ORC3/5 ubiquitylation in vivo (ORC3–9R and ORC5–7R versus WT proteins; Supplementary

Fig. 10b), suggesting a strong flexibility in the choice of the ubiquitin attachment sites by OBI1. Indeed, OBI1 overexpression stimulated the ubiquitylation of ORC3–9R and ORC5–7R (Supplementary Fig. 10c), showing that OBI1 could target other lysine residues. To obtain definitive non-ubiquitylable mutants, dominant negative experiments were then performed using lysine-less (0 K) versions of ORC3 and ORC5 (see Supplementary Fig. 10a) that were no longer ubiquitylated in vivo (Fig. 5a).

Functional characterisation of the lysine-less ORC3/5 variants showed that ORC3/5 0 K could bind to chromatin (Supplementary Fig. 10d), and associate with the ORC complex (Fig. 5b) and OBI1 (Supplementary Fig. 10e), like the WT counterparts. The unrelated protein EGFP did not displayed these properties. Moreover, the ORC3/5 0 K mutants did not perturb pre-RC formation (see Fig. 5d), as indicated by the similar amount of ORC1, ORC2 and the MCM2/7 complex on chromatin of control (Vector), WT ORC3/5, and ORC3/5 0K-expressing cells. As a functional control, expression of the CDT1 inhibitor geminin impaired MCM2/7 chromatin recruitment (Fig. 5d), as previously observed[29]. These results suggested that the conservative lysine-to-arginine substitutions introduced in ORC3/5 0 K do not affect the essential function of the ORC complex in origin licensing. However, ORC3/5 0 K expression resulted in defective DNA synthesis (BrdU fluorescence intensity) associated with accumulation of S-phase cells (Fig. 5c), as observed with *OBI1*-silenced cells (see Fig. 2b). In addition, cell fractionation revealed defective chromatin recruitment of proteins involved in DNA replication activation, such as DNA polymerase α, PSF1 and PCNA (Fig. 5d). Moreover, DNA stretching experiments revealed that inhibition of ORC3/5 ubiquitylation resulted in faster replication forks (Fig. 5e), as observed when origin firing is impaired (see Fig. 2d and Supplementary Fig. 6b). These results indicated that OBI1-catalysed ORC3/5 multi-mono-ubiquitylation per se is important for efficient origin activation.

## Discussion

The ubiquitin pathway is generally recognised as an important regulator of DNA replication. Indeed, during the licensing reaction, the pre-RC factors ORC1[18,19], CDC6[30], CDT1[31,32] and the licensing inhibitor geminin[29] are controlled by K48-linked poly-ubiquitylation and proteasomal degradation. Chromatin unloading of the replicative helicase at the end of replication involves K48-linked MCM7 poly-ubiquitylation[33,34]. Replicative DNA methylation maintenance relies on the action of the ubiquitin ligase UHRF1 to catalyse the mono-ubiquitylation of histone H3 lysine 23[35]. Here, we show that DNA replication origin activation is positively regulated by a novel ubiquitin signalling pathway that involves ORC3 and ORC5 multi-mono-ubiquitylation catalysed

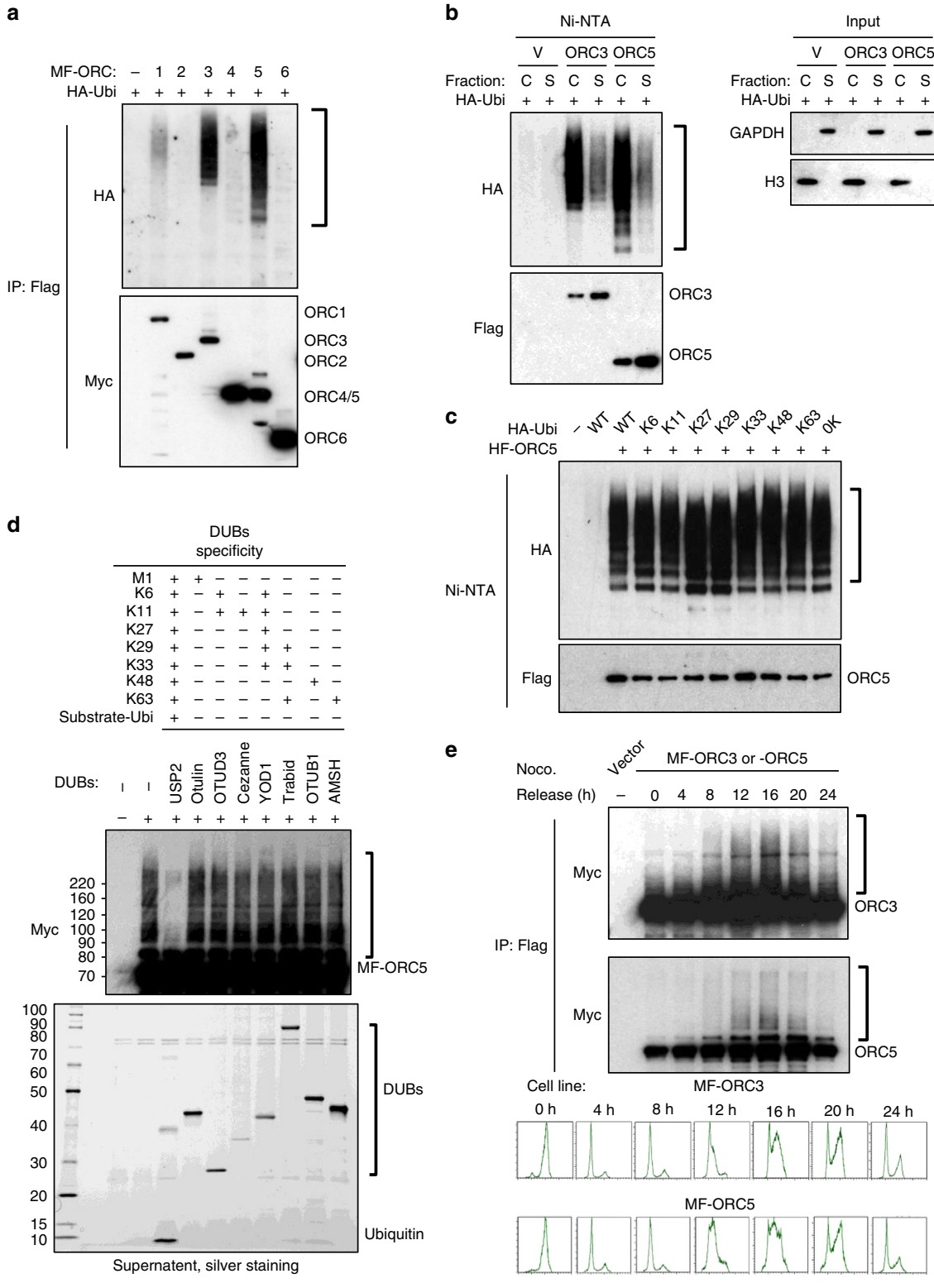

by OBI1 (C13ORF7/RNF219), a previously poorly characterised E3 ligase.

We found that OBI1 can interact with several ORC subunits (ORC1, ORC2, ORC3, ORC5 and ORCA/LRWD1, see Fig. 1c, e, Supplementary Fig. 3 and Supplementary Fig 10e), suggesting that it interacts with the whole complex. The fact that OBI1 ubiquitylates ORC3 and ORC5 in vitro also indicates that OBI1 can directly interact with these ORC subunits.

Replication origin selection is required for choosing the licensed origins to be activated amongst all potential origins and their timing of activation during S-phase. Our results suggest that both selection processes could be regulated by OBI1, as proposed in the following model (Fig. 6a). When cells exit mitosis and enter the G1-phase, ORC complexes catalyse the licensing of all potential origins, independently of OBI1 action, leading to the chromatin recruitment of the replicative helicase MCM2/7 in an

**Fig. 3** Chromatin-bound ORC3 and ORC5 are multi-mono-ubiquitylated in S-phase. **a** U2OS cells were co-transfected with the indicated Myc-Flag (MF)-tagged ORC subunits and HA-tagged ubiquitin. Two days post-transfection, cells were lysed in NEM-containing lysis buffer and tagged proteins were purified by anti-Flag immunoprecipitation. Expression and ubiquitin conjugation of the immunoprecipitates were monitored by western blotting, as indicated. **b** U2OS cells were co-transfected with the indicated $His_6$-Flag (HF)-tagged ORC subunits and HA-tagged ubiquitin. Chromatin (C) and soluble (S) fractions were isolated and ORC subunits were purified by nickel beads (Ni-NTA) under denaturing conditions. Purified proteins and input extracts were analysed by western blotting with antibodies against the indicated proteins. **c** U2OS cells were co-transfected with $His_6$-Flag-ORC5 (HF-ORC5) and the indicated HA-tagged ubiquitin single-lysine (K) mutants. In the 0 K mutant, all lysine residues were replaced by arginine residues. ORC5 ubiquitylation was detected by purification in denaturing conditions on nickel beads (Ni-NTA). Expression and ubiquitin conjugation of the isolated proteins were monitored by western blotting. **d** UbiCRest analysis of ubiquitylated ORC5. U2OS cells were transfected with Myc-Flag (MF)-tagged ORC5, without ectopic ubiquitin. Two days post-transfection, MF-ORC5 was immunoprecipitated (IP: Flag) and incubated with the indicated deubiquitylating enzymes (DUBs). Supernatants were recovered and analysed by silver staining. DUBs and released endogenous ubiquitin are indicated. Ubiquitylation was revealed by the presence of high molecular weight forms detected by western blotting against tagged ORC subunits. The ubiquitin linkage specificity of the used DUBs is indicated. **e** U2OS cells stably expressing Myc-Flag (MF)-tagged ORC3 or ORC5 were synchronised in mitosis by incubation with nocodazole and then released in normal growth medium for the indicated times. Tagged proteins were immunoprecipitated (IP: Flag) and ubiquitylation was assessed by anti-Myc antibody (upper panels). Synchronisation was evaluated by flow cytometry (lower panels)

inactive state. This is in agreement with the cell cycle-regulated OBI1 binding to chromatin and with the finding that OBI1 knockdown does not affect pre-RC formation. At S-phase onset, OBI1 could play the role of a replication origin selector involved in the selection of the origins to be fired by modifying a pool of replication origins through ORC multi-mono-ubiquitylation that would favour their preferential activation (Fig. 6b).

As it is well known that mono-ubiquitylation regulates protein/protein interactions[25], this modification might allow the recruitment of limiting firing factor(s) with ubiquitin-binding capacity to the pre-RCs to be fired. Mechanistically, multi-mono-ubiquitylation of the ORC complex may also alter the local chromatin environment around origins, making them more accessible to limiting firing factors. Consistent with this idea, it was recently reported that ORC5 interacts with the histone acetyl-transferase GCN5/KAT2A, making origins more accessible for activation[36]. ORC5 multi-mono-ubiquitylation could promote this interaction, opening the local origin chromatin environment and stimulating origin activation. Although we cannot exclude that other factors involved in replication origin activation might also be ubiquitylated by OBI1, ubiquitylation of ORC3/5 is essential because non-ubiquitylable ORC3/5 mimic OBI depletion effects on DNA replication. Importantly, our findings suggest that, in addition to its essential and well-established role in origin licensing during G1-phase, the ORC complex also has a second crucial role in origin activation in S-phase, as the substrate of OBI1 activity.

## Methods

**Reagents and antibodies**. The following reagents were purchased from Sigma-Aldrich: BrdU (B5002), IdU (I7125), N-ethylmaleimide (E3876), hydroxyurea (H8627), polybrene (H9268), puromycin (P9620), propidium iodide (P4864), Hoechst dye (H6024), anti-Flag coupled agarose beads (A2220), Flag peptide (F3290), anti-Flag (M8823), -Myc (M4439), -CDC6 (C0224), -cyclin A (C4710), -ORC1 (PLA0221), -MCM7 (M7931) and -PCNA (P8825) antibodies. CldU (105478) was from MP Biomedicals; magnetic affinity beads (Dynabeads M-450 goat anti-mouse) from Invitrogen; HA peptide from Roche; anti-HA coupled agarose beads (sc-7392 AC), anti-HA (sc-805), -ORC2 (sc-13238), -MCM2 (sc-10771), -ORC4 (sc-20634), -geminin (sc-74456; sc-74496), -OBI1/RNF219 (sc-84039), -RPA70 (sc-48425) and -CHK1 (sc-8408) antibodies were purchased from Santa-Cruz. Anti-RFC2 (ab88502), -histone H3 (ab62642), -RPA32 (ab76420), -polymerase alpha (ab176734), -CDC7 (ab10535), -$His_6$-tag (ab18184), -GAPDH (ab9484) and -PSF1 (ab181112) antibodies were from Abcam. Anti-BrdU and anti-IdU mouse IgG1 were from Becton Dickinson; anti-γ-H2AX (9718) and -phospho-CHK1 (2341) antibodies were from Cell Signaling; HRP-coupled anti-mouse, -rabbit and -goat antibodies from Amersham; TrueBlot HRP-coupled antibodies from Rockland; anti-CldU rat and anti-ssDNA mouse IgG2a antibodies from AbCys SA and Chemicon, respectively; anti-mouse FITC-coupled, anti-mouse IgG1 Alexa 546-coupled, anti-rat Alexa 488-coupled, anti-mouse IgG2a Alexa 647-coupled antibodies from Molecular Probes; anti-IL2Rα mouse monoclonal antibody from Millipore. Anti-CDT1 and anti-RNF219 rabbit polyclonal antibodies were generous gifts by M. Fujita and B. Sobhian, respectively. In-house anti-OBI1

rabbit polyclonal antibodies were generated against recombinant OBI1 N- and C-terminus. siRNAs against human OBI1 and ORC1 (SmartPool) and negative control (siGENOME Non-Targeting siRNA #2, D-001210–02) were purchased from Dharmacon. Boston Biochem provided the UbiCRest kit as well as E1, UbcH5, wild type and lysine-less ubiquitins and ubiquitin-aldehyde proteins. TNT reticulocyte in vitro translation kit was from Promega.

**Cell culture, infection and transfection**. T98G, HeLa S3, U2OS, NIH 3T3 and EcoPlatinum cells were cultured in DMEM supplemented with 10% foetal bovine serum (FBS), glutamine and antibiotics. HCT116 cells were grown in McCoy's 5 A medium supplemented with 10% FBS, glutamine and antibiotics. Plasmids (2–6 μg/6 cm dishes) and siRNAs (5–10 nM) were transfected using Lipofectamine (Invitrogen) or JetPEI (polyplus) and Interferin (Polyplus) reagents, respectively. To generate HeLa S3 cell lines that stably express tagged baits, cells were first transfected with a vector that expresses mouse cationic amino acid transporter-1 (mCAT1) to render them susceptible to ecotropic retroviruses. Two days after transfection, HeLa S3 cells were incubated with retroviruses produced in EcoPlatinum cells in the presence of 4 μg/mL Polybrene for 24 h. Transduced cells were selected using magnetic beads (Invitrogen) coupled to anti-IL2Rα antibodies and enriched populations were expanded. For cell transformation experiments, NIH 3T3 cells were co-transfected with OBI1-expressing plasmids (or empty vector as control) and the puromycin resistance vector (pBabe-puro). Two days after transfection, cells were selected in 2.5 μg/mL puromycin. To generate stable U2OS cell lines expressing Myc-Flag-ORC3 or –ORC5, cells were co-transfected with tagged-ORC expression plasmid along with a puromycin resistant vector in a ratio 10:1. Two days after transfection, cells were passaged in puromycin containing media. 2–3 days later, resistant transfected cells were seeded at low confluence and individual colonies were isolated, expanded and monitored for ectopic protein expression. Similarly, in dominant-negative experiments, U2OS cells were co-transfected with $His_6$-tagged ORC3/5 WT or 0 K (or the empty vector) and pBabe-puro. One day after transfection, cells were passaged in puromycin containing medium for two days. Selected cells were further cultured for 18–24 h without puromycin and processed for assays.

**Plasmid constructs**. The coding sequences of human ORC1, ORC2, LRWD1, CDC6 and CDT1 were cloned in the retroviral vector pOZ-FH-C[37]. For transient expression experiments, the described coding sequences were cloned in derivatives of the mammalian expression vector pCS3 to express N-terminally $Myc_6$-, $Myc_5$-Flag-, $His_6$- or Flag-tagged proteins, as indicated. OBI1 mutants were generated using PCR mutagenesis. EGFP and CDT1 expression vectors were previously described[38]. HA-tagged ubiquitin expression vectors were previously described[39]. The pBabe-puro retroviral vectors have been described[40]. Ubiquitin single-lysine mutants or lysine-less (0 K) expression vectors were obtained from Addgene. ORC3 and ORC5 lysine mutants were produced through gene synthesis (MWG). Details about DNA constructs available on request.

**Cell proliferation and transformation assays**. Proliferation was measured by counting viable cells after trypan blue staining each day for 4 days. The fold-increase in cell proliferation compared to day zero (siRNAs) is expressed as the mean of three independent experiments. The oncogenic properties of NIH 3T3 cells stably expressing OBI1 or empty vector were characterised by foci formation and soft agar growth assays, as previously described[38]. Briefly, for formation assays, stable populations were plated in 60 mm plates at equivalent cells density. Confluent cells were cultured for 2–3 weeks and the culture medium was changed every 2–3 days. Foci were stained with crystal violet dye (0.5% crystal violet, 4% formaldehyde, 30% ethanol and 0.17% NaCl). For quantification, the number of stained foci was counted from two independent experiments. For soft-agar growth, cells ($10^5$) were mixed with medium supplemented with 0.4% agarose and placed

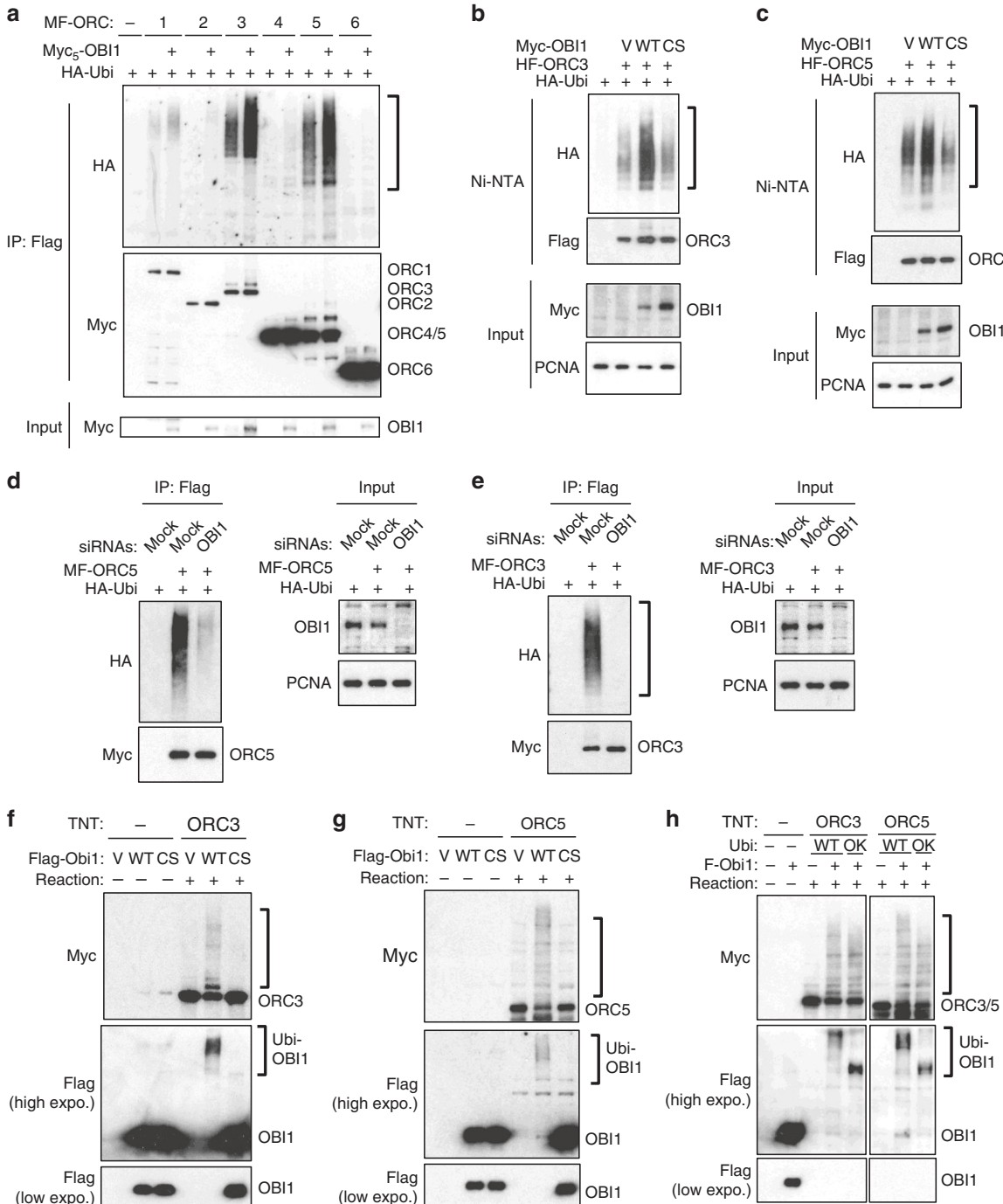

**Fig. 4** OBI1 catalyses ORC3 and ORC5 multi-mono-ubiquitylation. **a** U2OS cells were co-transfected with the indicated Myc-Flag (MF)-tagged ORC subunits and HA-tagged ubiquitin. When indicated ( + ), Myc-tagged OBI1 was also co-expressed. Two days post-transfection, cells were lysed in NEM-containing lysis buffer and tagged proteins were purified by anti-Flag immunoprecipitation. Expression and ubiquitin conjugation of the immunoprecipitates were monitored by western blotting, as indicated. Myc-OBI1 expression was also analysed in input extracts. **b, c** U2OS cells were co-transfected with $His_6$-Flag-ORC3 (HF-ORC3) or $His_6$-Flag-ORC5 (HF-ORC5) and either wild type (WT) or inactive (CS) Myc-OBI1 along with HA-ubiquitin as indicated. ORC3/5 ubiquitylation was detected by purification in denaturing conditions on nickel beads (Ni-NTA). Expression and ubiquitin conjugation of the isolated proteins were monitored by western blotting. Myc-OBI1 expression was also analysed in input extracts. **d, e** U2OS cells were transfected with Mock or *OBI1*-specific siRNAs. The next day, cells were co-transfected with Myc-Flag (MF)-tagged ORC3 or ORC5 and HA-tagged ubiquitin, as indicated. Ubiquitylation was detected 2 days later as in **a**. Expression of endogenous OBI1 was monitored in input extracts by western blotting. **f, g** OBI1 in vitro assay. U2OS cells were transfected with wild type Flag-OBI1 (WT) or inactive mutant (CS) for two days. Immunoprecipitated OBI1 (IP: Flag) was incubated with in vitro translated Myc-ORC3 or -ORC5 along with E1, E2 (UbcH5a), ubiquitin and ATP for 30 min. at 37 °C. Tagged proteins were detected by western blotting as indicated. **h** OBI1 in vitro assay performed as in **f, g**, but with wild type (WT) or lysine-less (0 K) ubiquitin. Tagged proteins were detected by western blotting as indicated

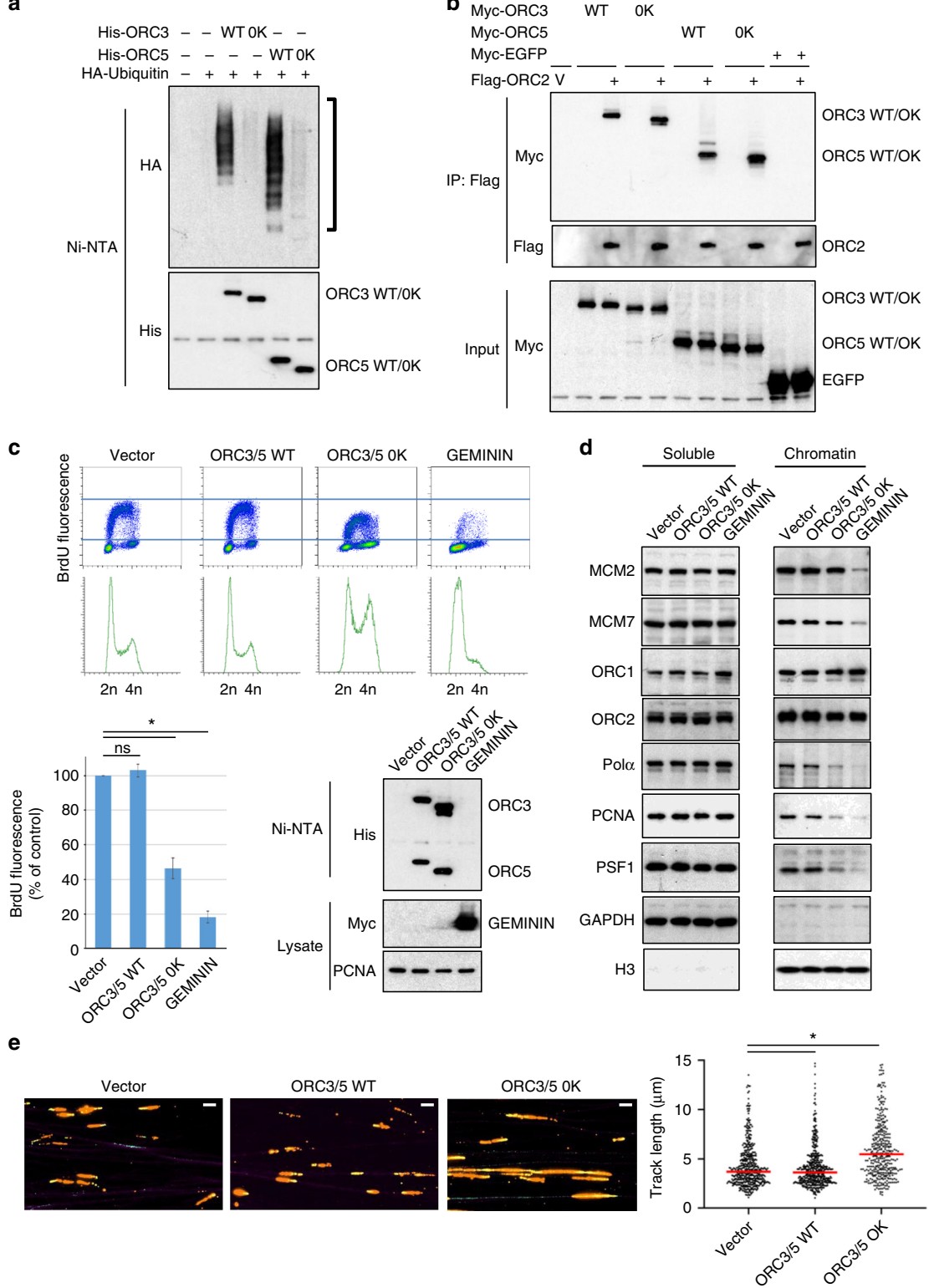

on top of the 1% agarose layer with growth medium. One millilitre medium was added to the solidified layer and changed every 2–3 days. After 3–4 weeks, colonies were photographed in randomly selected fields using a phase-contrast microscope.

**TAP-TAG purification.** Dignam nuclear extracts from 60 15-cm dishes of cells for each HeLa S3 cell line were obtained, as previously described[37,41]. Briefly, cells were collected and washed twice in ice-cold PBS. Cell pellets were resuspended in 10 cell pellet volume of ice-cold Hypotonic Buffer (HB: 10 mM Tris-HCl pH 7.3, 1.5 mM MgCl$_2$, 10 mM KCl, 1 mM DTT, 0.2 mM PMSF) and spun at 190 g at 4 °C for 5

min. Cell pellets were resuspended in two cell pellet volume of HB and incubated on ice for 10–15 min. for the cells to swell. Cells suspensions were transferred to an ice-cold Dounce, lysed with 10–15 strokes with the tight pestle and spun at 3000×g at 4 °C for 20 min to pellet nuclei. Nuclei were resuspended in half of the nuclei pellet volume (NPV) of ice-cold Low Salt Buffer (20 mM Tris-HCl pH 7.3, 1.5 mM MgCl$_2$, 0.2 mM EDTA, 20 mM KCl, 1 mM DTT, 0.3 mM PMSF). While gently vortexing, half of NPV of ice-cold High Salt Buffer (20 mM Tris-HCl pH 7.3, 1.5 mM MgCl$_2$, 0.2 mM EDTA, 1.2 M KCl, 1 mM DTT and 0.3 mM PMSF) was added dropwise. The extraction continued on a rotary wheel at 4 °C for 30 min. The suspensions were spun at 15,700 g at 4 °C in microcentrifuge tubes for 30 min. The

**Fig. 5** ORC3 and ORC5 multi-mono-ubiquitylation is essential for origin firing. **a** U2OS cells were co-transfected with His$_6$-tagged wild type (WT) ORC3 and ORC5 or lysine-less versions (0 K, see Supplementary Fig. 10a) and HA-ubiquitin, as indicated. Two days post-transfection, cells were lysed in denaturing conditions for His$_6$-tagged protein purification. Expression and ubiquitin conjugation of the isolated proteins were monitored by western blotting. **b** U2OS cells were co-transfected with Myc-tagged (WT or 0 K) ORC3 and ORC5 or EGFP and Flag-ORC2. Two days after transfection, cell lysates were Flag-immunoprecipitated to purify the ORC subunit, and the association of co-expressed proteins was analysed by western blotting. Expression of Myc-tagged proteins in input extracts was also monitored. **c** U2OS cells were transfected with ORC3 and ORC5 (WT or 0 K), Myc-geminin or empty vector and with a puromycin-resistance plasmid. The next day, transfected cells were selected in puromycin medium for 2 days, and grown for another day without selection. Cells were then pulsed with BrdU for 15 min and analysed by flow cytometry (upper panels). BrdU incorporation fluorescence signal was quantified from three independent experiments (lower left panel). Expression of ectopic proteins and endogenous PCNA was assessed by western blotting (lower right panels). ns, non-significant; *$p < 0.01$. **d** U2OS cells were treated as in **c**. Chromatin and soluble fractions were isolated and analysed by western blotting with antibodies against the indicated proteins. **e** DNA fibre stretching analysis. U2OS cells were treated as in **c**. Cells were incubated with IdU (15 min; red) followed by CldU (15 min; green) and processed for DNA spreading. Representative images are shown. Scale bars, 5 μm. The length of CldU tracks after the IdU signal was measured. More than 200 measurements are shown from two independent experiments. Red bars indicate median values. ns, non-significant; *$p < 0.01$

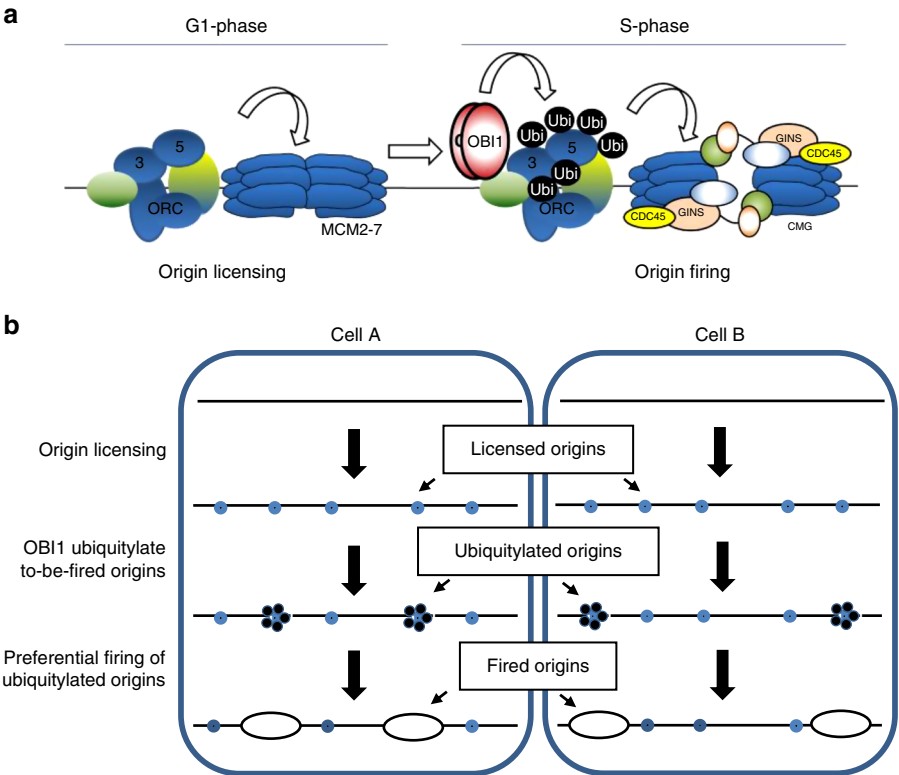

**Fig. 6** Proposed role of OBI1 in DNA replication origin activation. **a** In the G1-phase of the cell cycle, potential origins are licensed independently of OBI1 action, leading to the chromatin recruitment of the replicative helicase MCM2/7 in an inactive state. In S-phase, OBI1 catalyses the multi-mono-ubiquitylation of origin-bound ORC complexes, a modification that promotes origin firing. **b** OBI1 might select the origins to be activated in S-phase among all the potential origins licensed in G1 by marking them through ORC3 and ORC5 multi-mono-ubiquitylation. OBI1-selected pool of origins might differ from cell to cell, explaining the relatively low firing efficiency of mammalian origins. This modification of the ORC complex could act by recruiting putative ubiquitin-binding activation factors, or alternatively by opening the local chromatin environment, making it more accessible to limiting firing factors

supernatants (i.e. Dignam extracts) were frozen in liquid nitrogen at stored at −80 °C. TAP-TAG purification was performed as previously described[42,37]. Briefly, 8–10 mL of Dignam nuclear extracts were incubated with 80–100 μL of anti-Flag agarose beads at 4 °C for 4 h on a rotary wheel. After extensive washes in HA-IP buffer (20 mM Tris-HCl pH 7.5, 5 mM MgCl$_2$, 150 mM KCl, 10% glycerol, 0.1% Tween20, 0.05% NP-40, 1 mM DTT, 0.2 mM PMSF, 1 mM ATP), beads were eluted twice by competition using 200 ng/mL Flag peptide in 300 μL HA-IP buffer for 1 h at 4 °C. Pooled eluates were then incubated with 10 μL anti-HA beads at 4 °C for 2 h. After extensive washes in HA-IP buffer, complexes were eluted twice with 30 μL of the same buffer containing 400 ng/mL HA peptide for 1 h, as before. Purified complexes were then analysed by silver staining, western blotting or mass spectrometry.

**Mass spectrometry analysis**. Mass spectrometry (MS) analysis was performed at the Functional Proteomic Platform (IGF, Montpellier, France). TAP-TAG purified complexes were separated on NuPAGE gradient gels (Invitrogen) and stained using colloidal Coomassie (Biorad). Bands were excised, digested with trypsin (Promega) and processed for MS analysis. Samples (1 μl) were analysed online using a nanoflow HPLC system (Ultimate 3000, Dionex) coupled to a mass spectrometer with a nanoelectrospray source (LTQ-Orbitrap VELOS, Thermo Fisher Scientific). Peptides were separated on a capillary column (C18 reverse phase, Pepmap®, Dionex) with a gradient of 0–40% Buffer B in Buffer A for 150 min (A = 0.1% formic acid, 2% acetonitrile in water; B = 0.1% formic acid in 100% acetonitrile) at a flow rate of 300 nl/min. Spectral data were analysed using the Proteome Discoverer 1.2 software (Thermo Fisher Scientific) and Mascot (Matrix Science) version 2.3. The SwissProt CPS HUMAN databases, including the Cardation (M) modification, was used. The false discovery rate (FDR) was set at 1%. In Fig. 1b, interactions were visualised using Cytoscape 3.3.0 (cytoscape.org).

**Protein biochemistry**. To monitor protein expression, cells were lysed in Triton-lysis buffer (50 mM Tris-HCl pH 7.4, 100 mM NaCl, 5 mM EDTA, 40 mM β-glycero-phosphate, 50 mM NaF, 1% Triton X-100 and protease inhibitors), as

described previously[38]. In brief, cells were washed in ice-cold PBS and lysed in Triton buffer for 30 min. on ice. The crude lysates were spun at $15,700 \times g$ for 10 min. at 4 °C to obtain the final lysates. Protein concentration was evaluated using the BCA kit (Pierce). Soluble and chromatin fractions were obtained using the CSK buffer (10 mM PIPES pH 6.8, 100 mM NaCl, 300 mM Sucrose, 1 mM MgCl₂, 0.1–0.3% Triton X-100, 1 mM ATP, 0.5 mM DTT and protease inhibitors), as described previously[38]. In brief, cells were washed in ice-cold PBS and spun at $190 \times g$ for 5 min at 4 °C. Cell pellets were resuspended in 100–500 μL CSK buffer for 10 min on ice. Crude lysates were spun in microcentrifuge tubes at $1000 \times g$ at 4 °C for 3 min. The supernatants were collected and corresponded to the soluble fractions. The chromatin pellets were washed with the same volume of CSK buffer and incubated again 10 min. on ice. After another centrifugation, the supernatants were discarded, yielding the chromatin pellets. Both fractions were denatured in Laemmli sample buffer. For Flag-ORC/OBI1 co-immunoprecipitation experiments, cells were lysed in TK300 lysis buffer[43] (20 mM Tris-HCl pH 7.9, 300 mM KCl, 5 mM MgCl₂, 10% glycerol; 0.5% NP-40 plus protease inhibitors cocktail) on ice for 30 min. Equivalent amounts of protein samples were incubated at 4 °C with anti-Flag coupled agarose beads (10 μL) for 4 h. After extensive washing with TK300 lysis buffer, bound proteins were eluted in boiling Laemmli buffer. Samples were denatured in Laemmli sample buffer, ran on gradient gel (NuPAGE, Invitrogen) and transferred to nitrocellulose membrane. For western blot analysis, membranes were blocked with 5% milk in TBS/T buffer (20 mM Tris-HCl pH 7.6, 137 mM NaCl, 0.1% Tween20). Primary antibodies were diluted in 2% BSA in TBS/T buffer and incubated overnight with the membranes. Western blots were revealed using ECL Plus (Pierce) chemiluminescence reagent. The shown western blots are representative of two to three independent experiments. Silver staining was done using the SilverQuest silver staining kit (Invitrogen), following the recommended procedures.

Uncropped blots of main figures can be found in Supplementary Figs. 11, 12, 13 and 14.

**Ubiquitylation-related assays.** For native purification of ubiquitylated proteins, cells were lysed in Triton X-lysis buffer containing 10mM N-ethylmaleimide (NEM), as described previously[44]. Equal amounts of protein samples were incubated with anti-Flag agarose beads (10 μL) at 4 °C for 4 h. After extensive washes in lysis buffer, samples were analysed by western blotting. For purification of ubiquitylated proteins in denaturing conditions, cells were lysed in 500 μL of denaturing lysis buffer (100 mM NaH₂PO₄, 10 mM Tris base, 6 M guanidine hydrochloride, 10 mM imidazole, pH 8.0) for 30 min. on ice. Lysates viscosity was reduced by trituration pipetting. Clarified lysates were incubated with 10 μL Ni-NTA beads (Qiagen) at 4 °C for 2–3 h, and washed in denaturing washing buffer (100 mM NaH₂PO₄, 150 mM NaCl, 8 M urea, 22.5 mM imidazole, pH 8.0) 5 times. Samples were eluted in Laemmli buffer and analysed by western blotting. For purification of ORC3 and ORC5 from chromatin and soluble fractions, cells were processed using the CSK buffer as described above. Equivalent amounts of chromatin and solubles fractions (~100 μL) were denatured in denaturing lysis buffer (500 μL) and ORC subunits were purified as stated above. The UbiCRest assay was performed following the recommended conditions (Boston Biochem). Briefly, Myc-Flag-ORC3 and ORC5 transiently expressed in U2OS cells were purified in native conditions using 10 μL of Flag-beads, as described above. After extensive washes in lysis buffer followed by PBS, beads were incubated with a panel of deubiquitylases (DUBs) at 37 °C for 60 min. in supplied reaction buffer. Samples were vortexed and centrifuged to collect the supernatants. Beads and supernatants were analysed by western blotting and silver staining, respectively. To control for DUBs activity, Xenopus egg extract[45] were supplemented with recombinant HA-ubiquitin (200 ng/mL) and ATP (10 mM) for 15 min. at 30 °C. 10 μg of extracts were incubated with individual DUBs in 20 μL reaction buffer for 60 min. at 37 °C. Samples were denatured in Laemmli buffer and analysed by anti-HA western blot. For in vitro OBI1 ubiquitylation assay (based on)[46] Flag-OBI1 was immunoprecipitated from transiently transfected cells using 10 μL Flag-beads in TK300 lysis buffer supplemented with 2 mM DTT. After extensive washing in TK300 buffer, beads were washed twice with ubiquitylation reaction buffer (40 mM Tris-HCl pH 7.6, 5 mM MgCl₂, 10% glycerol, 1 mM DTT). Myc-tagged ORC3 and ORC5 cloned in pCS3 were in vitro translated using SP6 polymerase and cold aminoacids provided in the TNT kit as specified by the manufacturer. Purified OBI1 on beads was incubated with ORC substrates (1–3 μL), E1 (100 ng), UbcH5a (1 μM), wild type or lysine-less ubiquitin (1 mg/mL), ubiquitin-aldehyde (1 μM) and ATP (10 mM) in ubiquitylation reaction buffer (20 μL final) for 30 min. at 30 °C with agitation. Reactions were stopped with Laemmli buffer and analysed by western blotting.

**Flow cytometry.** For cell cycle analysis, ethanol-fixed cells were incubated with RNase A and propidium iodide, and analysed by flow cytometry using a MACS-Quant cytometer (Miltenyi) as described[38]. Alternatively, cells were pulsed with 100 nM BrdU for 15 min prior to fixing. After HCl denaturation, BrdU incorporation was detected using a mouse anti-BrdU antibody followed by an anti-mouse FITC-coupled antibody. BrdU incorporation was quantified by measuring the fluorescence intensity of BrdU-positive cells. The value of control cells (siMock) was normalised to 100%. The fluorescence value of BrdU-positive cells from experimental conditions was expressed as the percentage of fluorescence relative to

that of control cells. The summary graph is the mean of three independent experiments.

**DNA fibre analysis.** *Plugs preparation:* Asynchronous U2OS cells transfected with siRNAs for three days were labelled with two modified nucleosides: IdU and CldU. Cells were sequentially labelled with 30 μM IdU and 300 μM CldU for 20 min each, without intermediate washing. After labelling, cells were immediately placed on ice to stop DNA replication. Cells were then centrifuged (300 × g at 4 °C for 5 min) and washed three times with 1 × phosphate-buffered saline (PBS); 3 × 10⁶ cells were resuspended in 100 μl of 1 × PBS with 1% low-melting agarose to embed cells in agarose plugs. Plugs were incubated in 2 mL of proteinase K buffer (10 mM Tris-Cl, pH 7.0, 100 mM EDTA, 1% N-lauryl-sarcosyl and 2 mg/mL proteinase K) at 45 °C for 2 days (fresh solution was added on the second day).

*DNA molecular combing:* Digested proteins and other degradation products were completely removed by washing the plugs several times in TE₅₀ buffer (50 mM EDTA, 10 mM Tris-Cl, pH 7.0). Protein-free DNA plugs were then stored in TE₅₀ buffer at 4 °C, or used immediately for combing. Agarose plugs were stained with the YOYO-1 fluorescent dye (Molecular Probes) in TE₅₀ buffer for 1 h, washed few times with TE₅₀ buffer, resuspended in 100 μl of TE₅₀ buffer and melted at 65 °C for 15 min. The solution was maintained at 42 °C for 15 min, and then 10 units of β-agarase (Sigma Aldrich) was added overnight. After digestion, 4 ml of 50 mM MES (2-(N-morpholino) ethanesulfonic acid, pH 5.7) was added very gently to the DNA solution and then DNA fibres were combed and regularly stretched (2 kb/μm) on silanised coverslips, as described previously[47]. Combed DNA was fixed at 65 °C for at least 2 h, denatured in 1 N NaOH for 20 min, and washed several times in 1 × PBS. After denaturation, silanised coverslips with DNA fibres were blocked with 1% BSA and 0.1% Triton X100 in PBS. Immuno-detection was done with antibodies diluted in 1 × PBS, 0.1% Triton X100, 1% BSA and incubated at 37 °C in a humid chamber for 60 min. Each incubation step with antibodies was followed by extensive washes with 1 × PBS. Immuno-detection was done with an anti-ssDNA antibody (1/100 dilution, Merck Milipore), the mouse (1/20 dilution, clone B44 from Becton Dickinson) and rat anti-BrdU antibodies (1/20 dilution, clone BU1/75 Thermo Fisher Scientific) that recognise the IdU and CldU tracks, respectively. The mouse anti-BrdU antibody, clone B44, is derived from hybridisation of mouse Sp2/0-Ag14 myeloma cells with spleen cells from BALB/c mice immunised with iodouridine-conjugated ovalbumin. It reacts with iodouridine and BrdU[48,49]. The rat anti-BrdU antibody, clone BU1/75 (ICR1), cross-reacts with CldU, but not with thymidine or iododeoxyuridine[49]. The secondary antibodies were: goat anti-rat antibody coupled to Alexa 488 (1/50 dilution, Molecular Probes), goat anti-mouse IgG1 coupled to Alexa 546 (1/50 dilution, Molecular Probes), and goat anti-mouse IgG2a coupled to Alexa 647 (1/100 dilution, Molecular Probes). Coverslips were mounted with 20 μl of Prolong Gold Antifade (Molecular Probes), dried at room temperature for 12 h, and processed for image acquisition using a fully motorised Leica DM6000 microscope equipped with a CoolSNAP HQ2 1 CCD camera and controlled by MetaMorph (Roper Scientific). Images were acquired with a ×40 objective, where 1 pixel corresponds to 335 bp. As at the ×40 magnification, one microscope field of view corresponds to ~450 kb, observation of longer DNA fibres required the capture of adjacent fields. Inter-origin distances and replication fork speed were measured manually using the MetaMorph software. Statistical analyses of inter-origin distances and replication fork speed were performed using Prism 5.0 (GraphPad).

DNA spreading was performed as previously described[50–52]. Briefly, cells were pulsed with 20 μM iodo-deoxyuridine (IdU) for 15 min. followed by 200 μM chloro-deoxyuridine (CldU) for 15 min, trypsinised and counted. Two thousand cells were lysed in SDS spreading buffer (200 mM Tris-HCl pH 7.5, 50 mM EDTA, 0.5% SDS) and deposited on glass slides (StarFrost). The slides were tilted slightly and the drops were allowed to run down the slides slowly, then air dried, fixed in 3:1 methanol:acetic acid for 10 min, and allowed to dry. Glass slides were processed for immunostaining as for DNA combing analysis. Coverslips were mounted with Prolong Gold anti-fading agent. Images were acquired by immunofluorescence microscopy (Leica DM6000 or Zeiss ApoTome, RIO imaging facility) and a Coolsnap HQ CCD camera 25 (Photometrics), and analysed with the Metamorph software (Molecular Devices). Fork speed was measured in more than 200 forks using the ImageJ software.

**Statistical analysis.** No statistical methods were used to predetermine sample size. The experiments were not randomised, and investigators were not blinded to allocation during experiments and outcome assessment. Replication fork speed was estimated for individual forks with IdU track flanked by a CldU track. Only uninterrupted forks, as confirmed by DNA counterstaining, were analysed. Inter-origin distances were measured as the distance (kb) between the centres of two adjacent progressing forks located on the same DNA fibre. For the analysis of fork density, images were acquired in an unbiased manner (randomly), without first searching for the presence of IdU-CldU tracts. GraphPad Prism (GraphPad Software) and Excel (Microsoft) were used to generate graphs and perform statistical analyses. DNA replication parameters generally do not display a Gaussian distribution[53]. Therefore, distributions were compared using the non-parametric Mann–Whitney two-tailed test that does not assume a Gaussian distribution. For statistical significance, differences between two experimental groups were examined using Student's t-test. Statistical significance was set at *$P \leq 0.01$, **$P \leq 0.001$.

**Reporting summary**. Further information on research design is available in the Nature Research Reporting Summary linked to this article.

### Data availability
Data supporting the findings of this study are available within the article and its supplementary information files. All data are available from the corresponding authors upon reasonable request.

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

## Acknowledgements

We thank members of our laboratory for helpful suggestions. We acknowledge S. Urbach and E. Demettre (Functional Proteomic Platform) for the mass spectrometry analysis. We thank H. Tourrière and members of P. Pasero's Lab for reagents and technical advices. We are also grateful to A. Constantinou, M. Fujita and B. Sobhian for essential reagents. We acknowledge the Montpellier DNA Combing Facility for providing silanized coverslips. PC was supported by the 'Fondation pour la Recherche Médicale' (FRM) and the 'Fondation ARC'. J.N. was supported by The 'Ligue Nationale Contre le Cancer' (LNCC) and S.S. by FRM. Along the years, the research leading to these results has received funding from the European Research Council (ERC) under the European Community's Seventh Framework Programme (FP7/2007–2013 Grant Agreement no. 233339". This work was also supported by grants of the 'Agence Nationale de la Recherche' (ANR) to M.M. and Y.S., the ARC, the LNCC, and the FRM to M.M., the Centre Hospitalier Universitaire of Montpellier (Y.S.).

## Author contributions

P.C. and M.M. designed the experiments and wrote the paper; P.C. performed the experiments with J.N. and help from A.D.; S.S. performed the DNA combing analysis under the supervision of Y.S.; I.P. and S.B. contributed to cell culture, extracts preparation and antibodies production.

## Additional information

**Competing interests:** The authors declare no competing interests.

