## [Peer Review File · Nature Communications]

Reviewers' comments:

Reviewer #1 (Remarks to the Author):

The initiation of DNA replication requires the loading of Origin Recognition Complex (ORC) onto origins, followed by the assembly of Cdc6, Cdt1, and MCM2-7 to form the pre-replication complex (pre-RC). Once the origin is "licensed" by the loading of MCM2-7, the origin will fire during S phase. It is known that only a subset of origins will be "fired" during replication, yet the detailed mechanism which regulates the selection of "fired" origins remain to be completely understood.

In the current manuscript, the authors identified an E3 ligase ORC-ubiquitin-ligase-1 (OBI1, or RNF219), which interacts with ORC, and ubiquitinates several subunits of ORC (mainly Orc3 and Orc5). The depletion of OBI1 does not affect pre-RC establishment, but leads to defects in origin firing. The non-ubiquitylatable Orc3/5 mutants display impaired origin activation, and phenocopy OBI1-depletion, indicating that the ubiquitination of ORC is essential for origin firing. The authors provide solid evidence that the ubiquitination of Orc3 and Orc5 are essential for origin activation. Overall this is a solid manuscript with very clean and convincing high quality data. The data is significant and provides novel insights into the regulation of origin firing in mammalian cells. Some of the experiments that would make the data even more significant are suggested below, specifically inclusion of more details on how OBI1 ubiquitinates Orc3/5 would strengthen the manuscript.

Specific comments are listed below:

1. Is the interaction of OBI1 and ORC cell cycle regulated?
2. Is the ubiquitination of Orc3/5 by OBI1 cell cycle specific? What is the functional relevance of this ubiquitination if it is occurring throughout the cell cycle or perhaps occurring only during specific stages of the cell cycle.
3. Can the authors generate a mutant of Orc3/Orc5 that can no longer associate with OBI1 (or generate a mutant of OBI1 that can no longer interact with ORC and yet have an intact E3 ligase activity) and address if these mutants still display impaired origin activity. This would be the only way to demonstrate direct role of OBI1 in ubiquitinating ORC in vivo. Depletion experiments are supportive but not necessarily conclusive. What is the evidence that some other E3 ligase is not ubiquitinating Orc3/Orc5? Even if other E3 ligases are involved, finding OBI1 as an E3 ligase ubiquitinating ORC is significant.
4. As per Figure 3b, even though the most pronounced ubiquitination is for Orc3 and Orc5, there is marginal increase in ubiquitination for all ORC subunits. This again brings the question if the ubiquitination of other ORC subunits is cell cycle dependent.
5. Does OBI1 directly interact with Orc3/5, or does OBI1 interact with other ORC subunits and ubiquitinates Orc3/5? Some IP experiments are provided but direct interaction studies are not. It would be nice to include this at least as part of the discussion.
6. In addition to BrdU incorporation, how does the expression of Orc3/5 WT and OK affect fork speed, IOD, and GFD?

Reviewer #2 (Remarks to the Author):

In this manuscript the authors set it out to identify factors that may regulate the selective firing of replication origins in human cells – a question that has gone unanswered for quite some time. This goal is accomplished by the extensive, unbiased identification of proteins associated with components of the pre-RC by tandem-affinity purification/mass spectrometry approach. Numerous known components of the replication machinery are identified, validating the approach and a previously uncharacterized putative E3 ubiquitin ligase, RNF219, is detected. Depletion of this protein, which the authors cleverly dub OBI1, results in defective replication that appears to affect origin firing rather than licensing. The authors present data that suggests a model in which regulated chromatin-association of OBI1 catalyzed multi-monoubiquitylation of the pre-RC

components ORC3 and ORC5. The model further suggests that these modification events are required for the selective firing of a subset of replication origins, through mechanisms that will be worked out in future studies. The findings represent an important step forward the understanding of the origin firing program in higher eukaryotes and may have impact outside of this field. However, the study is not yet mature. The authors provide a strong cohort of data that are consistent with the proposed model, but often which often are open to other interpretations or do not stringently support the conclusions. Fortunately, data to strengthen support for the proposed model should be readily obtainable. My concerns for this intriguing study and potential strategies to address them are detailed below.

The authors provide detailed methods and have performed sufficient statistical analyses to allow other researchers to be confident in and able to reproduce this work.

Major Points:

1. It is not clear what is meant by Mock (siMock), but there is concern that this refers to a mock transfection rather than the use of control, non-targeting siRNA sequences. A mock transfection is not an adequate control as it does not introduce to cell physiology that general transfection of siRNA may have.

2. The authors should demonstrate that OBI1 is capable of directly ubiquitinating ORC3/5. Ideally this would be done with purified proteins in vitro. At the least, the authors should show that the interaction between the proteins is required for OBI1-mediated ubiquitination of ORC3/5. While the current data are consistent with OBI1 acting on the ORCs, it cannot be ruled out that it indirectly regulates ubiquitination of these proteins via another protein/mechanism.

3. To more definitively show that OBI1 is promoting the ubiquitinylation of ORCs, the experiments presented in Figs 3e, ED8a-b should be performed under denaturing conditions. While the data are consistent with increased ubiquitination caused by co-expression of OBI1, it are not conclusive. E3 ligases are known to undergo autoubiquitinylation. Thus, given that OBI1 robustly interacts with ORC3 and ORC5 (Fig ED1), probing for either MYC (both MYC-OBI1 and ORC subunits are MYC-tagged) or HA (for ubiquitin) in the FLAG-IPs cannot discriminate between ubiquitin-ORC conjugates and ORC-associated ubiquitin-OBI1 conjugates.

4. The model in which OBI1 promotes mutli-monoubiquitinylation of ORC3/5, as suggested by data in Figs 3b-c, ED9a, would benefit from further substantiation. The results of these studies are consistent with the interpretation that the ORCs undergo only monoubiquitinylation. However, given that these studies rely on the ability of the mutant ubiquitin to outcompete the endogenous ubiquitin in order to show an affect, the current data to not exclude the possibility that the Ub0K (or any of the other mutants) are simply not expressed at a high enough level to elicit an affect. To substantiate this model the authors should show that expression of Ub0K does impact the ubiquitin-conjugate pattern of a protein known to be polyubiquitinylated (e.g., p53). To further corroborate the model, the authors should show that inhibition of the proteasome does not impact the accumulation of the ORC3/5-Ub conjugates and that OBI1 does not impact the stability of these proteins, using cycloheximide assays for example. Similarly, the corroborating data from the UbiCREST assay would be bolstered by demonstration that these enzymes are indeed active.

5. The data in ED Fig 10 does not necessarily reflect OBI1-specific ubiquitinylation as concluded in the text as there is no manipulation of OBI1 activity and the modifications could result from the activity of another enzyme(s).

6. The data in ED Figs 4b-d should be quantified and additional images of the soft-agar assays provided. An image of 1 colony does not make a compelling case for an oncogenic potential. As it is this data tangential to the story and could also be removed.

7. The model that OBI1 regulates the firing of select origins is logical and consistent with the data, but requires additional support as there is no direct evidence to corroborate the major tenets of

the model. The model could be supported by several additional studies:

- a. The authors should demonstrate that ubiquitinylation of ORC3/5 happens on the chromatin-associated pool and is temporally correlated with S-phase.
- b. The authors should show that only a pool of chromatin-associated ORC3/5 is ubiquitinated. This data may perhaps be obtained by showing that depletion of the ORC3/5-Ub conjugates (e.g., by HA immunoprecipitation) from chromatin fractions does not deplete the entire pool, for example. Additionally, the authors may demonstrate that OBI1 and Ubiquitin are associated with only a population of ORC one sights in on the chromatin – perhaps using an immunofluorescence approach.
- c. It is not clear whether the decrease in BrdU incorporation in the ORC3/5 OK-expressing cells is due to failure to fire or diminished elongation. DNA combing analysis may help here. Additionally, the authors may consider a knock-down/replacement strategy with the OK mutants that may better demonstrate the requirement of ORC3/5 ubiquitinylation in the absence of wild-type proteins.

Minor Points:

1. The identity of “Loading” in ED-Fig3b should be provided.
2. p-values for the ONCOMINE data in ED-4a should be provided.
3. Page 5 second paragraph the sentence, “To study in detail this previously unrecognized post-translational modification...” It would be more accurate to use uncharacterized rather than unrecognized, which implies that ubiquitinylation of ORC3/5 has no previously been observed.
4. The blot in ED Fig 9b is mislabeled. MF-ORC3 is present where indicated not to be and USP2 does not have an effect on Ub-conjugates as stated in the main text.

Answers to Reviewers:

Reviewer #1:

We thank Reviewer #1 for her/his positive review and constructive comments. Like the second reviewer, Reviewer #1 recognizes that the selection process of to-be fired DNA replication origins amongst all potential ones is poorly understood. We strongly believe that our description of OBI1 function provides important insights into this process. We wish to highlight the fact that the Reviewer evaluated the manuscript's data to be 'very clean', 'convincing' and of 'high quality'. Also, Reviewer #1 judged that our study 'provided solid evidence that the ubiquitination of Orc3 and Orc5 are essential for origin activation'. Here is a detailed response to Reviewer #1's comments. Additional experiments were performed and the new results strengthen our initial conclusion on the positive role of OBI1 in origin firing through ORC3/5 multi-mono-ubiquitylation.

Comments 1 and 2:

Is the interaction of OBI1 and ORC cell cycle regulated? Is the ubiquitination of Orc3/5 by OBI1 cell cycle specific? What is the functional relevance of this ubiquitination if it is occurring throughout the cell cycle or perhaps occurring only during specific stages of the cell cycle.

Reviewer #1 wonders if the activity of OBI1 on the ORC complex could be cell cycle regulated. This is an important information that could help us understand how ORC multi-mono-ubiquitylation and origin firing are linked. To answer this question directly, we monitored ORC3/5 ubiquitylation during the cell cycle, because this modification is the end-product of OBI1 activity. We generated stable U2OS cell lines expressing tagged-ORC3/5. These cells were synchronized in mitosis and released in the cell cycle for different time points. Our results revealed that ORC3/5 ubiquitylation is cell cycle regulated (**New Fig 3e**). Specifically, ORC3/5 ubiquitylation was low during mitosis and early G1-phase, started to increase in late G1-/early S-phase, peaked in S-phase, and decreased in G2/M-phase cells. These results show that ORC3/5 ubiquitylation occurred concomitantly with origin firing. Also, these results are consistent with our model proposing that OBI1-catalyzed ORC ubiquitylation is involved in the selection of the origins to be activated.

Comment 3:

Can the authors generate a mutant of Orc3/Orc5 that can no longer associate with OBI1 (or generate a mutant of OBI1 that can no longer interact with ORC and yet have an intact E3 ligase activity) and address if these mutants still display impaired origin activity. This would be the only way to demonstrate direct role of OBI1 in ubiquitinating ORC in vivo. Depletion experiments are supportive but not necessarily conclusive. What is the evidence that some other E3 ligase is not ubiquitinating Orc3/Orc5? Even if other E3 ligases are involved, finding OBI1 as an E3 ligase ubiquitinating ORC is significant.

Reviewer #1 wants us to generate OBI1/ORC interaction mutants and test their activity. Also, she/he wonders whether other ubiquitin ligases could be involved in ORC3/5 ubiquitylation. First, we generated several OBI1 deletion mutants and tested their association with the ORC complex. We found that OBI1 C-terminal and coiled-coil domains are important for its ability to interact with the ORC complex (**New ED Fig 3a**). Conversely, OBI1 ubiquitin ligase activity

is not required for ORC interaction (**New ED Fig 3a and 3d**). We then tested the effect of these mutants on ORC ubiquitylation. We found that OBI1 mutants that cannot associate with ORC are also unable to induce ORC5 ubiquitylation (**New ED Fig 9c**). These results provide further evidence that OBI1 is an ORC3/5 ubiquitin ligase. We cannot formally exclude the possibility that other E3 could be involved in ORC3/5 ubiquitylation, depending on the physiological and cellular context; this possibility is now mentioned in the revised manuscript (**page 9**). Nevertheless, we believe that the provided data clearly show that OBI1 is a major ORC3/5 ubiquitin ligase in vivo. Indeed, siRNA-mediated knockdown experiments showed a strong loss of ORC3/5 ubiquitylation in OBI1-depleted cells (Fig 4d-e).

Comment 4:

As per Figure 3b, even though the most pronounced ubiquitination is for Orc3 and Orc5, there is marginal increase in ubiquitination for all ORC subunits. This again brings the question if the ubiquitination of other ORC subunits is cell cycle dependent.

Reviewer #1 wonders about the possibility that OBI1 could ubiquitylate other ORC complex subunits in addition to ORC3/5. First, we found that ORC3/5 were much more ubiquitylated compared with other ORC subunits (Fig 3a). ORC1 also was ubiquitylated, but to a lesser extent, and it is known that ORC1 ubiquitylation occurs in S-phase and is mediated by the SCF^{SKP2} E3 ligase (Mendez J (2002) Mol Cell 9:481). In our gain of function experiments, we found that OBI1 stimulated ORC3 and ORC5 ubiquitylation, while leaving other ORC subunits largely unaffected (Fig 4a), in agreement with the conclusion that the main activity of OBI1 is for ORC3/5. We are commenting this point in the revised manuscript (**pages 8-9**).

Comment 5:

Does OBI1 directly interact with Orc3/5, or does OBI1 interact with other ORC subunits and ubiquitinates Orc3/5? Some IP experiments are provided but direct interaction studies are not. It would be nice to include this at least as part of the discussion.

Answering this question would require extensive in vitro and structural studies that we believe to be beyond the scope of the present initial identification and characterization of OBI1, a still poorly characterized protein. However, new in vitro OBI1 ubiquitylation assays, now provided in the revised manuscript, show that OBI1 can recognize and ubiquitylate ORC3 and ORC5 alone (**New Fig 4f-h**). The manuscript provides several experiments showing that OBI1 interacts with several subunits of the ORC complex (ORC1, ORC2, ORC3, ORC5 and ORCA) (Fig 1c, 1e and ED Fig 3, 10e), consistent with the idea that OBI1 can associate with the whole ORC complex. The way OBI1 recognizes the ORC complex is now further discussed in the revised manuscript (**page 11**).

Comment 6:

In addition to BrdU incorporation, how does the expression of Orc3/5 WT and OK affect fork speed, IOD, and GFD?

Reviewer #1 wished us to study the effect of non-ubiquitylatable ORC3/5 by DNA fiber analysis. It is well established that the reduction of DNA initiation events (caused by licensing or firing defects) leads to faster replication forks, as a compensatory mechanism (Montagnoli A (2008) Nat Chem Biol 4:357; Boos D (2013) Science 340:981; Zhong Y (2013) J Cell Biol

201:373; Kumagai A (2017) Mol Biol Cell 28:2998; Cottineau J (2017) J Clin Invest 127:1991). In agreement, we observed that ORC1 or CDC7 knockdown resulted in higher fork speed using DNA combing and DNA stretching analyses (Fig 2d and ED Fig 6b). OBI1 depletion also induced a similar fork speed phenotype (Fig 2d and ED Fig 6b). To answer the Reviewer's comment, we performed new DNA stretching experiments in cells expressing ORC3/5-OK and found that ORC3/5 ubiquitylation inhibition led to faster fork speed (**New Fig 5e**). Together with our BrdU incorporation and cell fractionation experiments (Figure 5c-d), these results are consistent with a role for ORC3/5 ubiquitylation in origin firing, thus mimicking OBI1 depletion.

Reviewer #2

We would like to thank Reviewer #2 for her/his positive and constructive comments. Like reviewer #1, Reviewer #2 acknowledges that the mechanism of origin selection is an important open question in the DNA replication field. She/He believes that our *'findings represent an important step forward in the understanding of the origin firing program in higher eukaryotes'*. We took into account the reviewer's comments, and we believe that our initial conclusions on the positive role of OBI1 in origin firing through ORC3/5 multi-mono-ubiquitylation are now strengthened in the revised manuscript. Here are our detailed responses to the Reviewer's inquiries.

Comment 1:

It is not clear what is meant by Mock (siMock), but there is concern that this refers to a mock transfection rather than the use of control, non-targeting siRNA sequences. A mock transfection is not an adequate control as it does not introduce to cell physiology that general transfection of siRNA may have.

In siRNA-mediated silencing experiments, Reviewer #2 asked about the exact meaning of our siMock conditions. In the manuscript, siMock corresponded to cells transfected with a non-targeting siRNA. We apologize for the confusion. We have incorporated the non-targeting sequence in the **new Supplementary Table 3**. A description of the 'siMock' condition is also incorporated in the revised manuscript (**page 5**).

Comment 2:

The authors should demonstrate that OBI1 is capable of directly ubiquitylating ORC3/5. Ideally this would be done with purified proteins in vitro. At the least, the authors should show that the interaction between the proteins is required for OBI1-mediated ubiquitination of ORC3/5. While the current data are consistent with OBI1 acting on the ORCs, it cannot be ruled out that it indirectly regulates ubiquitination of these proteins via another protein/mechanism.

Reviewer #2 wished us to test whether OBI1 could directly ubiquitylate ORC3/5 in vitro. In addition, she/he wished us to show that the interaction between OBI1 and ORC is required for ubiquitylation. To answer this comment, we performed new OBI1 in vitro ubiquitylation experiments. In these assays, tagged OBI1 was immunoprecipitated from transiently transfected cells, while the substrates ORC3/5 were in vitro translated in rabbit reticulocyte

lysates. We observed robust ubiquitylation of ORC3/5 catalyzed by OBI1 in vitro (**New Fig 4f-g**). Conversely, we did not detect any ubiquitylation when using the OBI1 RING mutant (CS). This shows that the observed activity was catalyzed by OBI1 and not by a contaminating ubiquitin ligase. Of note, in these conditions, OBI1 catalyzed multi-mono-ubiquitylation of ORC3/5 (**New Fig 4h**, see Comment 4).

In a second new series of experiments, we generated several OBI1 deletion mutants and evaluated their ORC binding and ubiquitylation activity. We found that OBI1 C-terminal and coiled-coil domains are required for ORC interaction by co-immunoprecipitation experiments (**New ED Fig. 3a**). Conversely, OBI1 ubiquitin ligase activity was not involved in ORC binding (**New ED Fig. 3a and 3d**). Gain of function experiments revealed that OBI1 association with the ORC complex is essential for its ability to ubiquitylate ORC5 (**New ED Fig. 9c**). We believe that these new results strengthen our initial conclusion that OBI1 is an E3 ubiquitin ligase for ORC3/5.

Comment 3:

To more definitively show that OBI1 is promoting the ubiquitylation of ORCs, the experiments presented in Figs 3e, ED8a-b should be performed under denaturing conditions. While the data are consistent with increased ubiquitination caused by co-expression of OBI1, they are not conclusive. E3 ligases are known to undergo autoubiquitylation. Thus, given that OBI1 robustly interacts with ORC3 and ORC5 (Fig ED1), probing for either MYC (both MYC-OBI1 and ORC subunits are MYC-tagged) or HA (for ubiquitin) in the FLAG-IPs cannot discriminate between ubiquitin-ORC conjugates and ORC-associated ubiquitin-OBI1 conjugates.

In the initial manuscript, ORC3/5 ubiquitylation was analyzed under native and denaturing conditions (see Fig 3a and 3c for example), demonstrating that the proteins are genuinely ubiquitylated in vivo (as observed in large-scale ubiquitylome studies (Wagner SA (2011) Mol Cell Prot 10:M111013284; Kim W (2011) Mol Cell 44:325). As noted by the reviewer, our OBI1 overexpression experiments were performed in native conditions only, leaving the possibility that the observed increased ubiquitylation signal could come from associated OBI1. In the revised manuscript, these gain of function experiments were performed under denaturing conditions, as suggested by the reviewer. These experiments demonstrated unambiguously that OBI1 overexpression stimulated ORC3/5 ubiquitylation (**New Fig 4b-c**). This effect required OBI1 ubiquitin ligase activity (**New Fig 4b-c**).

Comment 4:

The model in which OBI1 promotes multi-monoubiquitylation of ORC3/5, as suggested by data in Figs 3b-c, ED9a, would benefit from further substantiation. The results of these studies are consistent with the interpretation that the ORCs undergo only monoubiquitylation. However, given that these studies rely on the ability of the mutant ubiquitin to outcompete the endogenous ubiquitin in order to show an affect, the current data do not exclude the possibility that the Ub0K (or any of the other mutants) are simply not expressed at a high enough level to elicit an affect. To substantiate this model the authors should show that expression of Ub0K does impact the ubiquitin-conjugate pattern of a protein known to be polyubiquitylated (e.g., p53). To further corroborate the model, the authors should show that inhibition of the proteasome does not impact the accumulation of the ORC3/5-Ub conjugates and that OBI1 does not impact the stability of these proteins,

using cycloheximide assays for example. Similarly, the corroborating data from the UbiCREST assay would be bolstered by demonstration that these enzymes are indeed active.

Reviewer #2 wished us to further substantiate our conclusion that ORC3/5 are subjected to multi-mono-ubiquitylation. This conclusion is based on two independent assays, namely utilization of ubiquitin variants in which specific lysine residues were mutated, and of ubiquitin linkage specific DUBs (UbiCRest assay, Fig 3c-d and ED Fig 8a-b). To directly show that OBI1 catalyzed ORC3/5 multi-mono-ubiquitylation, we performed in vitro OBI1 ubiquitylation assay using lysine-less (OK) ubiquitin, which is unable to form any poly-ubiquitin chain, or wild type (WT) ubiquitin. These experiments showed a very similar pattern of ORC3/5 ubiquitylation between WT and OK ubiquitin (**New Fig 4h**). These in vitro OBI1 assays directly show that ORC3/5 were multi-mono-ubiquitylated, as revealed by our prior in vivo analysis. Nevertheless, the UbiCRest assay was further characterized, as requested by the reviewer. Particularly, Reviewer #2 wonders whether the linkage specific DUBs were active. To test this, we incubated DUBs with extracts expressing HA-tagged ubiquitin, expected to display a repertoire of ubiquitin linkage types. The universal DUB USP2_{CD} efficiently removed HA-ubiquitin conjugation (**New ED Fig 8c**). Also, we observed partial digestion using several linkage-specific DUBs (**New ED Fig 8c**), potentially reflecting the prevalence of their substrates. These results show that the enzymes supplied with the UbiCRest kit display activity. Altogether, we believe that our new results strongly support our initial conclusion that ORC3/5 are subjected to multi-mono-ubiquitylation catalyzed by OBI1.

Comment 5:

The data in ED Fig 10 does not necessarily reflect OBI1-specific ubiquitylation as concluded in the text as there is no manipulation of OBI1 activity and the modifications could result from the activity of another enzyme(s).

To generate non-ubiquitylable versions of ORC3/5, we first mutated lysine residues found ubiquitylated in large-scale ubiquitylome studies (ED Fig 10a). Ubiquitylation of ORC3-9R and ORC5-7R was found to be very similar to the of their wild type counterpart (ED Fig 10b). These results suggest that OBI1 can modify several other lysine side-chains. Based on the molecular weight of multi-mono-ubiquitylated ORC3/5 (see Fig 3d), we estimated that as many as 20 ubiquitin molecules can be conjugated to the ORC proteins. Thus, ORC3-9R and ORC5-7R should be still ubiquitylated on other lysine residues by OBI1. Reviewer #2 wished us to confirm this possibility. In gain of function experiments, OBI1 overexpression stimulated ORC3-9R and ORC5-7R ubiquitylation (**New Fig 10c**), demonstrating that these proteins are OBI1 substrates. This result shows that OBI1 can modify numerous sites on ORC3/5, further justifying our lysine-less approach to abrogate ORC3/5 ubiquitylation.

Comment 6:

The data in ED Figs 4b-d should be quantified and additional images of the soft-agar assays provided. An image of 1 colony does not make a compelling case for an oncogenic potential. As it is this data tangential to the story and could also be removed.

As requested by the reviewer, quantification and additional images were added to the figure describing OBI1 oncogenic properties (**new ED Fig 4b-d**).

Comment 7a:

The model that OBI1 regulates the firing of select origins is logical and consistent with the data, but requires additional support as there is no direct evidence to corroborate the major tenets of the model. The model could be supported by several additional studies. The authors should demonstrate that ubiquitylation of ORC3/5 happens on the chromatin-associated pool and is temporally correlated with S-phase.

To substantiate our model on OBI1 role in origin firing, the reviewer suggested looking whether ORC3/5 ubiquitylation occurred on chromatin. Also, Reviewer #2 wonders about the cell cycle regulation of ORC3/5 ubiquitylation. We answered to these two suggestions. First, we analyzed the ubiquitylation status of ORC3/5 found in chromatin and soluble fractions. These experiments showed that ORC3/5 ubiquitylation occurs mainly on chromatin (**New Fig 3b**). Second, we analyzed ORC3/5 ubiquitylation status during the cell cycle. For this, we generated stable U2OS cell lines expressing tagged ORC3 or ORC5 (without HA-ubiquitin). These cells were synchronized in mitosis by nocodazole treatment and released into the cell cycle at different time points. Tagged ORC3/5 were immunoprecipitated and their ubiquitylation was assessed by the appearance of high molecular weight species (which consist of ubiquitin conjugates, as shown by the UbiCrest assay, see Fig 3d and ED Fig 8b). We observed that ORC3/5 ubiquitylation is low in mitotic and early G1-phase cells (**New Fig 3e**). Notably, ORC3/5 ubiquitylation was induced in late G1/early S-phases, peaked in S-phase, and decreased as cells finished the cell cycle. This analysis clearly showed that ORC3/5 ubiquitylation occurred concomitantly with origin firing. Altogether, these results are consistent with our model stating that OBI1 ubiquitylates chromatin-bound ORC3/5 during S-phase, favoring their activation.

Comment 7b:

The authors should show that only a pool of chromatin-associated ORC3/5 is ubiquitylated. This data may perhaps be obtained by showing that depletion of the ORC3/5-Ub conjugates (e.g., by HA immunoprecipitation) from chromatin fractions does not deplete the entire pool, for example. Additionally, the authors may demonstrate that OBI1 and Ubiquitin are associated with only a population of ORC one sights in on the chromatin – perhaps using an immunofluorescence approach.

If OBI1 were involved in the selection of to-be-fired origins, only a pool of ORC3/5 should be ubiquitylated. Reviewer #2 thus wished us to evaluate the stoichiometry of ORC3/5 ubiquitylation. In our experiments related to the cell cycle regulation of ORC3/5 ubiquitylation (**New Fig 3e**), both unmodified and ubiquitin-conjugated ORC were detected, allowing us to assess ORC3/5 ubiquitylation stoichiometry. We estimated that a small proportion (5-10%) of ORC3/5 became modified. This pool of ubiquitylated ORC3/5 could represent the fraction associated with activated origins. This is now commented in the revised manuscript (**pages 8 and 11**).

Comment 7c:

It is not clear whether the decrease in BrdU incorporation in the ORC3/5 OK-expressing cells is due to failure to fire or diminished elongation. DNA combing analysis may help here. Additionally, the authors may consider a knock-down/replacement strategy with the OK

mutants that may better demonstrate the requirement of ORC3/5 ubiquitylation in the absence of wild-type proteins.

Reviewer #2 wished us to further characterize the DNA synthesis defects in cells expressing non-ubiquitylable forms of ORC3/5 by DNA fiber analysis. It is established that DNA initiation event reduction (caused by licensing or firing defects) induces faster replication fork as a compensatory mechanism (Montagnoli A (2008) Nat Chem Biol 4:357; Boos D (2013) Science 340:981; Zhong Y (2013) J Cell Biol 201:373; Kumagai A (2017) Mol Biol Cell 28:2998; Cottineau J (2017) J Clin Invest 127:1991). Indeed, we observed that ORC1 or CDC7 knockdown resulted in higher fork speed, as indicated by our DNA combing and DNA stretching analyses (Fig 2d and ED Fig 6b). OBI1 depletion also induced a similar fork speed phenotype (Fig 2d and ED Fig 6b). DNA stretching analysis of cells overexpressing wild type or lysine-less ORC3/5 or empty vector revealed that ORC3/5-OK did not impede elongation. In fact, we observed faster replication forks in ORC3/5-OK cells (**New Fig 5e**), as observed in OBI1-silenced cells. Together with our BrdU incorporation and cell fractionation experiments (Fig 5c-d), these results are consistent with the notion that ORC3/5 ubiquitylation per se is important for origin firing, thus mimicking OBI1 depletion.

Minor Point 1:

The identity of “Loading” in ED-Fig3b should be provided.

The loading was a non-specific band. Input loading is now controlled using an anti-PCNA antibody (**New ED Fig 3b**).

Minor Point 2:

p-values for the ONCOMINE data in ED-4a should be provided.

P-values for the ONCOMINE data are now provided in the revised figure (**new ED Fig 4a**).

Minor Point 3:

Page 5 second paragraph the sentence, “To study in detail this previously unrecognized post-translational modification...” It would be more accurate to use uncharacterized rather than unrecognized, which implies that ubiquitylation of ORC3/5 has not previously been observed.

We agree with Reviewer #2 that our initial sentence regarding ORC3/5 ubiquitylation was confusing. The revised manuscript now states that ORC3/5 ubiquitylation is a previously uncharacterized post-translation modification (**page 7**).

Minor Point 4:

The blot in ED Fig 9b is mislabeled. MF-ORC3 is present where indicated not to be and USP2 does not have an effect on Ub-conjugates as stated in the main text.

As observed by the Reviewer, our initial figure depicting the UbiCRest analysis of tagged ORC3 was mislabeled. We apologize for this error originating from re-scaling the panel during figure assembly. In the new Figure, labels are now well aligned with the appropriate lanes, showing that only USP2_{CD} can efficiently deubiquitylate tagged ORC3 (**New ED Fig 8b**).

Finally, as requested by the reviewer, we provide more details regarding methods and statistical analysis in the revised Methods section.

REVIEWERS' COMMENTS:

Reviewer #1 (Remarks to the Author):

The authors have satisfactorily addressed all my prior concerns. As I also mentioned in my prior review, this is a solid manuscript with very clean and convincing high quality data. The authors provide convincing data supporting the role of OBI1 in origin firing through ORC3/5 multi-mono-ubiquitination. I am impressed with the way the authors have performed additional relevant experiments and addressed all the comments. This will be an excellent addition to Nature Communications. A few minor corrections/suggestions are noted below:

1. Page 5, line 124 OBI1 is misspelled as OI1.
2. The Figure Extended 4C is self-explanatory, but it would be good to have the quantification for Extended Figure 4C.
3. The ORC3/5 OR mutant may be behaving as a dominant negative mutant since the WT counterparts are still present in the cell. This could be mentioned in the text.

Supriya Prasanth

Reviewer #2 (Remarks to the Author):

In this revised manuscript the authors have addressed all concerns of the reviewers from the initial submission and have incorporated suggested experiments to strengthen the discovery of the regulation of origin firing by OBI1-mediated ubiquitination of ORCs 3/5. The manuscript is suitable for publication in Nature Communications.

Answers to reviewers.

Reviewer1 :

The authors have satisfactorily addressed all my prior concerns. As I also mentioned in my prior review, this is a solid manuscript with very clean and convincing high quality data. The authors provide convincing data supporting the role of OBI1 in origin firing through ORC3/5 multi-mono-ubiquitination. I am impressed with the way the authors have performed additional relevant experiments and addressed all the comments. This will be an excellent addition to Nature Communications. A few minor corrections/suggestions are noted below:

1. Page 5, line 124 OBI1 is misspelled as OI1.

Correction done.

2. The Figure Extended 4C is self-explanatory, but it would be good to have the quantification for Extended Figure 4C.

In this supplementary Figure, we show that Obi1 overexpression resulted in colonies in soft-agar. With the control NIH 3T3 cells transformed by the vector, we never observed colonies in all our plates of soft agar (Supplementary Fig.4c, upper 4 panels). These colonies were only observed in our plates overexpressing Obi1 (Supplementary Fig.4c, lower 4 panels). We showed in the corresponding panels representative microscopic fields that we believe clearly show that Obi1 has transforming activity, also confirmed in Figure 4b by foci formation assays. In the legend of the extended figure, we have now added: "Panels show representative microscopic fields of the cell plates used in these experiments"

3. The ORC3/5 OR mutant may be behaving as a dominant negative mutant since the WT counterparts are still present in the cell. This could be mentioned in the text.

It is now mentioned in the text: "Here, the ORC3/5 OK mutant may be behaving as a dominant negative mutant."

Reviewer 2

In this revised manuscript the authors have addressed all concerns of the reviewers from the initial submission and have incorporated suggested experiments to strengthen the discovery of the regulation of origin firing by OBI1-mediated ubiquitination of ORCs 3/5. The manuscript is suitable for publication in Nature Communications.